# Prostaglandin EP2 receptor downstream of Notch signaling inhibits differentiation of human skeletal muscle progenitors in differentiation conditions

Fusako Sakai-Takemura[1], Ken'ichiro Nogami[1,2], Ahmed Elhussieny[1,3], Kota Kawabata[1], Yusuke Maruyama[1], Naohiro Hashimoto[4], Shin'ichi Takeda[1] & Yuko Miyagoe-Suzuki[1✉]

Understanding the signaling pathways that regulate proliferation and differentiation of muscle progenitors is essential for successful cell transplantation for treatment of Duchenne muscular dystrophy. Here, we report that a γ-secretase inhibitor, DAPT (N-[N-(3,5-difluorophenacetyl-L-alanyl)]-S-phenylglycine tertial butyl ester), which inhibits the release of NICD (Notch intercellular domain), promotes the fusion of human muscle progenitors in vitro and improves their engraftment in the tibialis anterior muscle of immune-deficient mice. Gene expression analysis revealed that DAPT severely down-regulates *PTGER2*, which encodes prostaglandin (PG) E2 receptor 2 (EP2), in human muscle progenitors in the differentiation condition. Functional analysis suggested that Notch signaling inhibits differentiation and promotes self-renewal of human muscle progenitors via PGE2/EP2 signaling in a cAMP/PKA-independent manner.

[1] Department of Molecular Therapy, National Institute of Neuroscience, National Center of Neurology and Psychiatry, Tokyo, Japan. [2] Department of Neurology, Neurological Institute, Graduate School of Medical Sciences, Kyushu University, Fukuoka, Japan. [3] Department of Neurology, Faculty of Medicine, Minia University, Minia, Egypt. [4] Department of Regenerative Medicine, National Institute for Longevity Sciences, National Center for Geriatrics and Gerontology, Aichi, Japan. ✉email: miyagoe@ncnp.go.jp

Duchenne muscular dystrophy (DMD) is a devastating muscle disease caused by mutations of the *DMD* gene, which encodes dystrophin. Currently there is no effective treatment for DMD[1]. Transplantation of muscle progenitors/precursors is a therapeutic strategy for DMD[2]. However, clinical trials of myoblast transfer in the 1990s were all unsuccessful. Experiments using mouse models suggested that the majority of transplanted myoblasts were lost immediately after transplantation[3–5].

Human induced pluripotent stem cells (hiPSCs) can be induced to differentiate into skeletal muscle cells even after extensive expansion[6–10]. Therefore, hiPS cells are expected to provide sufficient amounts of muscle progenitors for cell therapy. Recently, we reported an improved sphere culture-based protocol for induction of muscle progenitors from hiPSCs[10]. Induced muscle progenitors efficiently formed multinucleated myotubes in vitro and differentiated into myofibers in immune-deficient dystrophin-deficient *mdx* mice. However, the number of dystrophin-positive myofibers in *mdx* muscle was not satisfactory[10], requiring further investigation to clarify why myogenic cells, which differentiate efficiently into myotubes in vitro, do not form myofibers in vivo after engraftment.

Notch is a key regulator of myogenesis during development and postnatal life[11–15]. Recently, Low et al. reported that Dll4 activates Notch3 to regulate self-renewal in mouse C2C12 cells and mouse primary myoblasts[16]. Baghdadi et al. revealed that Notch keeps the satellite cells in their niche partly via collagen V-calcitonin receptor signaling[17]. These reports using mouse models emphasize again that Notch is indispensable for generation and maintenance of muscle satellite cells. On the other hand, the effects of Notch activation on engraftment remain controversial. Parker et al. reported that activation of Notch signaling during ex vivo expansion enhanced the efficiency of engraftment in a canine-to-murine xenotransplantation model[18]. In contrast, Sakai et al. reported that mouse muscle stem cells and human myoblasts treated with Notch ligands in vitro restored PAX7 expression but did not improve regeneration capacity after transplantation into mice[19].

Here, we report that a γ-secretase inhibitor, DAPT (N-[N-(3,5-difluorophenacetyl-L-alanyl)]-S-phenylglycine tert. butyl ester), which blocks Notch signaling, stimulates differentiation of human myogenic cells, mainly via blockage of prostaglandin E2/EP2 receptor signaling, and improves cell transplantation efficiency. We also show that COX-2/PGE2/EP2 signaling promotes self-renewal of human muscle progenitors via cAMP/PKA-independent signaling pathways.

## Results

### A Notch inhibitor, DAPT, promoted myotube formation by human muscle progenitors.
First, to explicate the effects of Notch signaling on differentiation of human muscle progenitors, we added DAPT, which specifically inhibits the γ-secretase complex and, as a result, blocks Notch signaling (Fig. 1a), to the cultures of human muscle progenitors. DAPT increased both the fusion index and myotube diameter of Hu5/KD3 cells, a human muscle progenitor cell line[20] (Fig. 1b–e), hiPS-derived myogenic cells (Fig. 1f–i), and adult human primary myoblasts (Supplementary Fig. 1), suggesting that Notch inhibition stimulated the recruitment of hiPS-derived muscle progenitors and postnatal myogenic cells, which otherwise do not fuse, to fusion.

### DAPT improved engraftment of human muscle progenitors.
Next, we tested whether DAPT improves engraftment of human muscle progenitors by promoting differentiation of engrafted cells (Fig. 2). DAPT was added to a suspension of Hu5/KD3 cells just before transplantation into pre-injured tibialis anterior (TA) muscles of immunodeficient NOD/Scid mice. Interestingly, DAPT improved the efficiency of cell transplantation of Hu5/KD3 cells (Fig. 2a–c). We then tested whether DAPT improved the efficiency of cell transplantation of hiPSC-derived muscle progenitors. DAPT was added to the cell suspension just before direct injection into the TA muscle of NSG-*mdx*[4CV] mice, then injected into the engrafted TA muscle four times with 2-day intervals (Fig. 2d). DAPT treatment increased the numbers of human lamin A/C-positive dystrophin-positive myofibers (Fig. 2d–f).

### Identification of Notch signal-responsive genes in human muscle progenitors in differentiation conditions.
To clarify the Notch target genes that inhibit or augment myotube formation, we examined the gene expression in Hu5/KD3 human myoblasts treated with DAPT for 4 days using RNA-seq analysis (Fig. 3a, b). We found that relatively limited numbers of the genes were up- (60 genes) or downregulated (67 genes) more than twofold by DAPT (Fig. 3b). In addition to protein-coding mRNA, 14 non-coding RNAs were found to be differentially expressed (Fig. 3b). We list the 10 most upregulated and the 10 most downregulated genes after DAPT treatment in Table 1. *NOTCH3* and two well-known Notch target genes, *HES1* and *HEY1*, were listed as the most downregulated genes, confirming that DAPT successfully inhibited Notch signaling. In the RNA-seq analysis, *NOTCH4* mRNA was extremely low in both groups. *NOTCH1* expression was also low, and it was not up- or downregulated with DAPT treatment (Supplementary DATA). Among Notch ligands, only *JAG1* was differentially expressed (downregulated) by DAPT treatment (Supplementary DATA).

Except LINC00948 (Linc-RAM) and IGFN1, most of the genes listed in Table 1 as "upregulated" have not been reported to be involved in skeletal muscle differentiation. Linc-RAM is a recently identified long non-coding RNA, which codes a SERCA activity-regulating small molecule, MYOREGULIN. Full-length Linc-RAM, but not Myoregulin, is reported to enhance myogenic differentiation in mice[21]. IGFN1 is also reported to be required for fusion of C2C12 myoblasts[22].

Next, by qPCR, we confirmed the downregulation of *NOTCH3*, *HEY1*, *HEYL*, *PTGER2(EP2)*, *COL6A3*, *APOE*, *CMKLR1*, *UNC5B*, and *SCG2*, and upregulation of *ID1* in Hu5/KD3 cells treated with DAPT (Fig. 3c, Supplementary Fig. 2). There was no significant difference between the expression levels of *NOTCH2* in DAPT- and non-treated cells at day 4. The expression of *NOTCH3* was very low at day 0 and gradually increased in control cells, but the increase was not observed in DAPT-treated cells. The expression of *PTGER2* was transiently increased at day 4. In contrast, the expression of *PTGER2* was gradually decreased in DAPT-treated cells (Fig. 3c). The same expression patterns were observed in human primary myoblasts (Supplementary Fig. 3).

### NOTCH3 and EP2 were highly expressed in self-renewing human muscle progenitors.
Next, we examined NOTCH3 expression in differentiation of Hu5/KD3 cells by FACS. NOTCH3 expression was detected in a fraction of the cells in high cell-density culture, but not on proliferating cells cultured at a low density (Supplementary Fig. 4b). The induction was completely abolished by DAPT treatment (Supplementary Fig. 4c). To clarify the function of NOTCH3, NOTCH3-high, and NOTCH3-negative cells were sorted by FACS (Fig. 4b). Then, total RNA was extracted from halves of these two fractions to perform qPCR. The other halves of the cells were plated onto collagen-I-coated plates at nearly confluency.

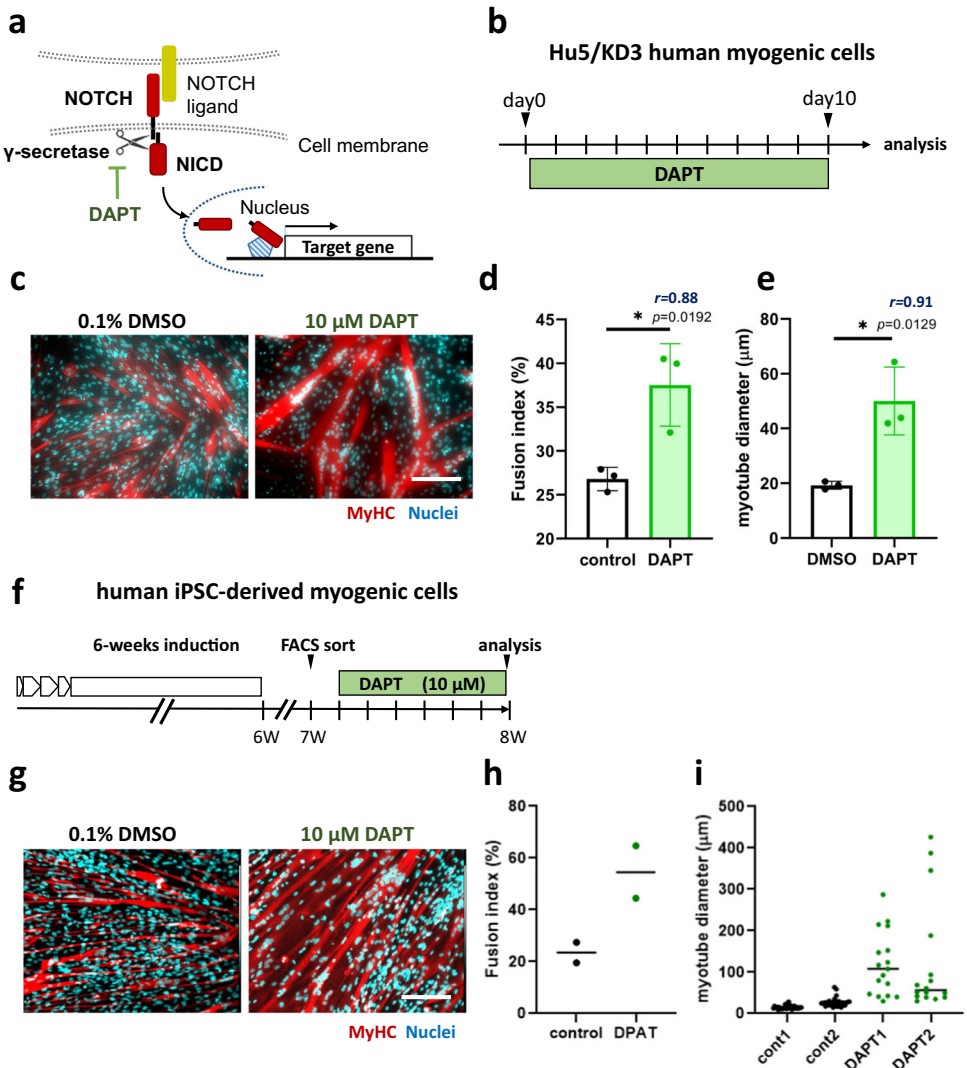

**Fig. 1 γ-secretase inhibitor DAPT promoted differentiation of hiPSC-derived muscle progenitors. a** DAPT inhibited Notch signaling by inhibiting γ-secretase. **b** Experimental design-1. Hu5/KD3 cells were plated onto collagen-I-coated plates and cultured for 10 days in 10% FBS/DMEM with or without DAPT, and the fusion index was determined at day 10. **c** Representative photos of myotube formation by Hu5/KD3 cells with or without DAPT. **d** Quantification of fusion index in **c**. Data are expressed as dot plot in control (0.1% DMSO treatment) and DAPT (10 μM DAPT treatment) cells. Data were analyzed by unpaired two-tailed Student's *t*-test. *n* = 3 samples/group. **e** Quantification of myotube diameter in (**c**). More than 10 myotubes were measured per group. Data were analyzed by an unpaired two-tailed Student's *t*-test. In **d**, **e**, the effect size of Pearson's *r* correlation (*r*) is shown. **f** Experimental design 2. After 6 weeks of sphere culture-based muscle induction, cells were plated onto collagen-I-coated plates and cultured for 7 days in 10% FBS/DMEM. Then ERBB3(+)CD271(+) cells as muscle progenitors were sorted by FACS. Sorted cells were cultured for a further 6 days with (10 μM) or without DAPT (0.1% DMSO). Induction and sorting of muscle progenitors was performed twice (2 experiments per group). **g** Representative photos of myotubes formed by hiPSC-derived ERBB3(+)CD271(+) cells with or without DAPT. **h** Quantification of fusion index in **g**. Three wells per sample were examined. The average of each sample was shown as dot. **i** Distribution of myotube diameter in **g**. Diameter of more than 15 myotubes per sample were measured. In **c** and **g**, myotubes were stained with an antibody against skeletal muscle myosin (MF20, red) and DAPI (Nuclei, blue). Scale bars, 200 μm.

*NOTCH2*, *NOTCH3*, *HEY1*, *HES1*, *HEYL*, *EP2*, and *EP4* (Fig. 4d) were expressed in NOTCH3-high cells at higher levels than in NOTCH3-negative cells. As expected, NOTCH3-negative cells robustly and rapidly formed myotubes, but NOTCH3-positive cells hardly formed myotubes or proliferated during 3 days culture (Fig. 4c), suggesting that NOTCH3 signal inhibits muscle differentiation as reported by Low et al.[16].

**Prostaglandin E2/EP2 signal inhibited differentiation of human myogenic progenitors.** Recently, Notch signaling was reported to keep mouse muscle satellite cells quiescent by reciprocal Notch-collagen V-calcitonin receptor signaling[17]. Interestingly, our RNA-seq showed that DAPT downregulated the collagens *COL1A1*, *COL1A2*, *COL3A1*, *COL4A1*, *COL4A2*, *COL5A1*, *COL5A2*, *COL5A3*, *COL6A1*, *COL6A2*, *COL6A3*, and *COL8A1* (Supplementary DATA) and G-protein-coupled prostaglandin E2 receptor 2 (EP2) (Table 1), which activates adenylate cyclase to convert ATP to cAMP, like calcitonin receptor. This similarity prompted us to examine the roles of PGE2/EP2 signaling in regulation of differentiation of muscle progenitors. Importantly, the NOTCH3-high fraction expressed PTGER2 (av. 9.0-fold) and PTGER4 (av. 3.8-fold) at higher levels than NOTCH3-negative cells (Fig. 4d). Although the results suggested

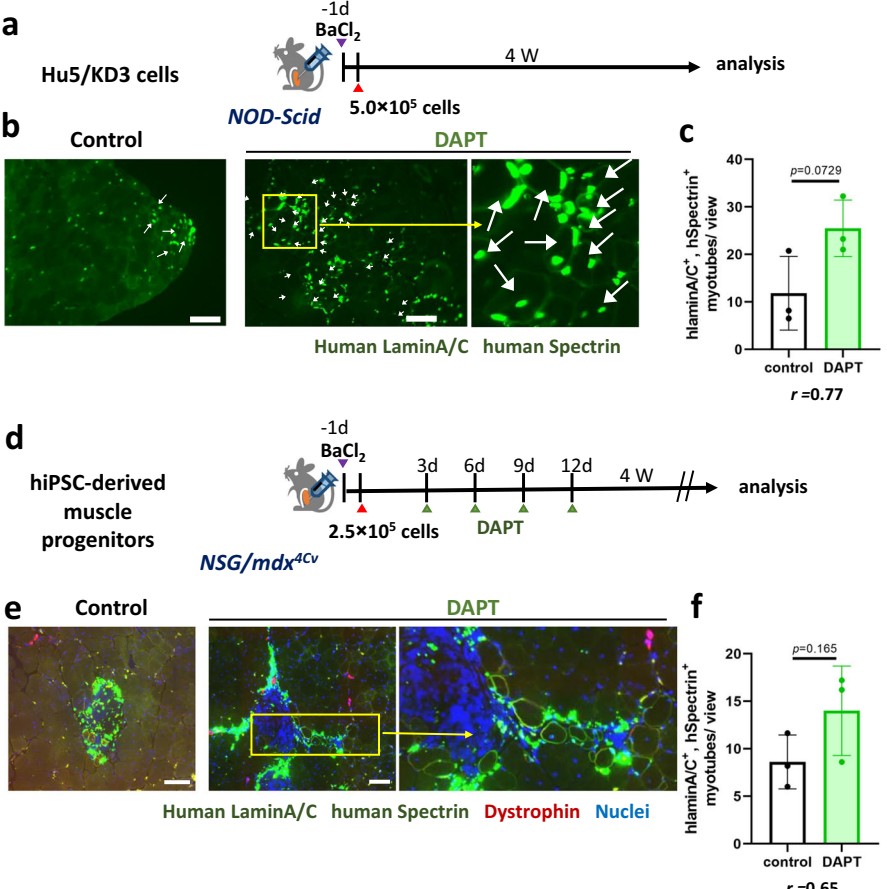

**Fig. 2 Notch inhibitor DAPT improved transplantation efficiency. a** Experimental design-1. To evoke muscle regeneration, $BaCl_2$ was injected into TA muscles of *NOD-scid* mice 24 h before transplantation. The next day, Hu5/KD3 cells ($5.0 \times 10^5$ cells) were transplanted into damaged TA muscles with or without DAPT. TA muscles were isolated 4 weeks after transplantation. **b** Engraftment and differentiation of a human myoblast cell line, Hu5/KD3 cells, with or without DAPT. Donor cell-derived myofibers were detected as human lamin A/C (nuclear membrane)-positive and human spectrin (plasma membrane)-positive myofibers. Scale bar = 100 μm. **c** The number of human lamin A/C- and human spectrin-positive myofibers per view. Data were analyzed by unpaired two-tailed Student's *t*-test. *n* = 3 mice/group. For each mouse, 4–8 muscle sections were examined. The effect size of Pearson's *r* correlation (*r*) is shown. **d** Experimental design 2. hiPSC-derived muscle progenitors (ERBB3(+)CD271(+) cells) were transplanted into TA muscles of *NSG-mdx^4Cv* mice with or without DAPT. $BaCl_2$ injection and sampling of TA muscles was performed as in (**a**). DAPT was injected into the engrafted TA muscle four times every 3 days after transplantation. **e** Representative photos of engraftment and differentiation of human iPSC-derived DAPT-treated muscle progenitors. **f** The number of human lamin A/C- and human spectrin-positive myofibers per view. Data were analyzed by unpaired two-tailed Student's *t*-test. *n* = 3 mice/group. For each mouse, 4–16 slices were examined. The effect size of Pearson's *r* correlation (*r*) is also shown. In **b**, **e**, scale bar indicates 100 μm.

a strong relationship between NOTCH3 and prostaglandin E2 (PGE2) signaling, addition of PGE2 did not upregulate NOTCH3 (Supplementary Fig. 4c), negating a positive-feedback from EP2 signaling to NOTCH3 expression. To further examine the relationship between NOTCH3 and EP2, we overexpressed NOTCH3 NICD (NICD3) in Hu5/KD3 cells. NICD3 did not increase EP2 expression, suggesting that NOTCH3 is not upstream of EP2 (Fig. 4e). As expected, overexpression of NICD3 completely prevented Hu5/KD3 cells from differentiation into myotubes (Supplementary Fig. 4d–f).

EP2 was widely expressed on mononuclear cells, but the expression level was reduced in DAPT-treated mononuclear cells (Fig. 5b–c). To directly examine the roles of EP2 signaling in muscle differentiation, we added prostaglandin E2 (PGE2) to the differentiating culture of Hu5/KD3 cells (Fig. 5d). PGE2 reduced the fusion index in a dose-dependent manner (Fig. 5d, e), but muscle differentiation activity was quickly recovered after washout of PGE2 (Supplementary Fig. 5a–c). An EP2-specific agonist, butaprost, also increased the percentage of mononuclear

cells (Fig. 5d, f). In contrast, TG6-10-1[23], an EP2 receptor antagonist, promoted fusion of muscle progenitors (Fig. 5d, g). We also examined the effects of overexpression of EP2 on differentiation of Hu5/KD3 cells by using a plasmid vector expressing either EP2-ires-DsRed or EP2-GFP fusion protein. Upregulation of EP2 blocked the differentiation of Hu5/KD3 cells (Fig. 6a–d, Supplementary Fig. 5d–f). In contrast, knockdown of EP2 improved fusion of muscle progenitors (Fig. 6e–i). These results suggest that the PGE2/EP2 signal inhibits differentiation of human muscle progenitors in differentiation conditions.

**Blockage of EP2 signaling promoted differentiation of iPSC-derived myogenic cells.** Next, we examined the effects of activation and blockage of EP2 receptor on hiPSC-derived muscle progenitors (Fig. 7). As expected, DAPT and TG6-10-1, an antagonist of EP2, greatly improved the fusion of the cells to the same degree. Unexpectedly, PGE2 and butaprost, an agonist of EP2, had no significant effects on the fusion index (Fig. 7), probably because differentiation of hiPSC-derived myogenic

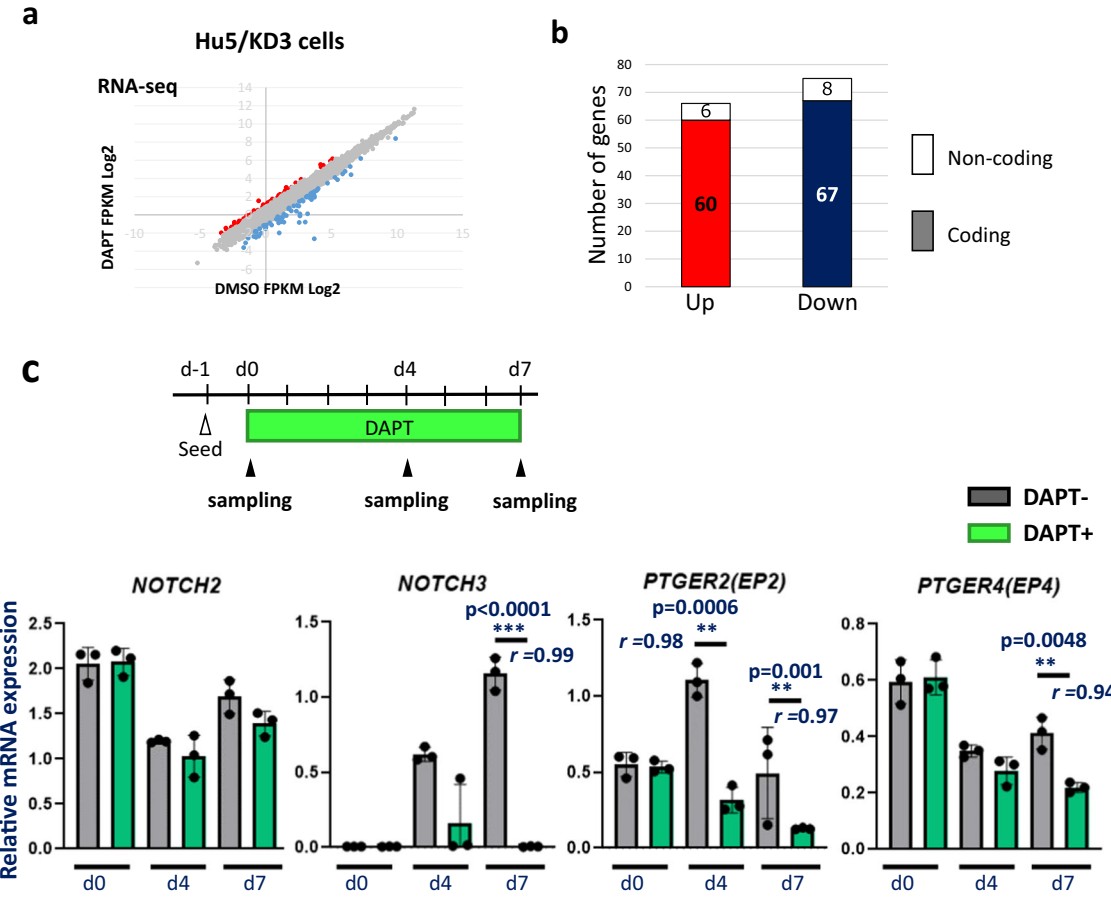

**Fig. 3 Identification of Notch target genes in muscle progenitors in differentiation conditions. a** Scatter graph of gene expression levels in differentiating Hu5/KD3 cells with (y-axis) or without DAPT treatment (x-axis). RNA was extracted from the differentiating cells after 4 days DAPT treatment and analyzed by RNA-seq. FPKM: fragments per kilobase of exons per million reads mapped. **b** Numbers of genes differentially (>2-fold) expressed in differentiating Hu5/KD3 cells with or without DAPT treatment. **c** Experimental design and qPCR analysis of NOTCH2, NOTCH3, PTGER2, and PTGER4 in Hu5/KD3 cells with or without DAPT treatment. RNA was extracted from the cells at day 0, day 4, and day 7. $n = 3$ samples/group. Data are shown as mean ± S.D. and analyzed by unpaired two-tailed Student's t-test. The effect size of Pearson's r correlation (r) is shown.

**Table 1 (related to Fig. 3) List of genes up- (top 10 genes) or downregulated (top 10 genes) in Hu5/KD3 cells by DAPT treatment.**

| Upregulated (top 10 genes) | | Downregulated (top 10 genes) | |
|---|---|---|---|
| Gene name | Fold change♯ | Gene name | Fold change♯ |
| ID1 | 2.1 | NOTCH3 | −6.3 |
| PLCL1 | 1.6 | HEY1 | −3.8 |
| MROH7 | 1.6 | HES1 | −3.4 |
| MRLN | 1.4 | PTGER2 | −3.1 |
| LINC00948 | 1.4 | COL6A3 | −3 |
| IGFN1 | 1.4 | APOE | −2.7 |
| RAB11FIP4 | 1.4 | CMKLR1 | −2.5 |
| IQGAP2 | 1.4 | UNC5B | −2.2 |
| ADAMTSL3 | 1.4 | SCG2 | −2.1 |
| IFI6 | 1.4 | CCDC102B | −1.9 |

♯DAPT FPKM Log2-DMSO FPKM Log2

**Contribution of EP4 to self-renewal of human myogenic cells.** Prostaglandin E2 has four seven-transmembrane G protein-coupled receptors (GCPRs), EP1–4. Among them, EP2 and EP4 activate adenyl cyclase, but the two receptors have different structures and functions. Therefore, we tested the effects of an EP4 antagonist, ONO-AE3-208, on the differentiation of myogenic cells. ONO-AE3-208 did not increase the fusion index of Hu5/KD3 myogenic cells (Fig. 8d), suggesting that the contribution of EP4 to self-renewal of myogenic progenitors is small.

**PGE2 produced by COX-2 activated EP2 signaling.** Next, we examined the upstream of EP2 using COX inhibitors. Indomethacin, which inhibits both COX-1 and COX-2, improved the fusion of Hu5/KD3 cells and hiPSC-derived myogenic progenitors (Fig. 8b, Supplementary Fig. 6d). COX-1-selective SC-560 did not promote differentiation of Hu5/KD3 cells and hiPSC-derived muscle progenitors (Fig. 8c, Supplementary Fig. 6f). In contrast, the COX-2-selective valdecoxib promoted the differentiation of Hu5/KD3 cells and hiPSC-derived muscle progenitors (Fig. 8c, Supplementary Fig. 6e), suggesting that COX-2 is involved mainly in EP2-mediated suppression of muscle differentiation.

We measured the concentration of PGE2 by an enzyme-linked immunosorbent assay (ELISA). Unexpectedly, the PGE2 level was quite low in the culture medium of hiPSCs-derived myogenic

cells is severely suppressed by TGF-β signaling, as previously reported[10]. DAPT slightly suppressed the proliferation of hiPSC-derived muscle progenitors, but there was no significant difference in cell numbers among the other groups (Fig. 7d).

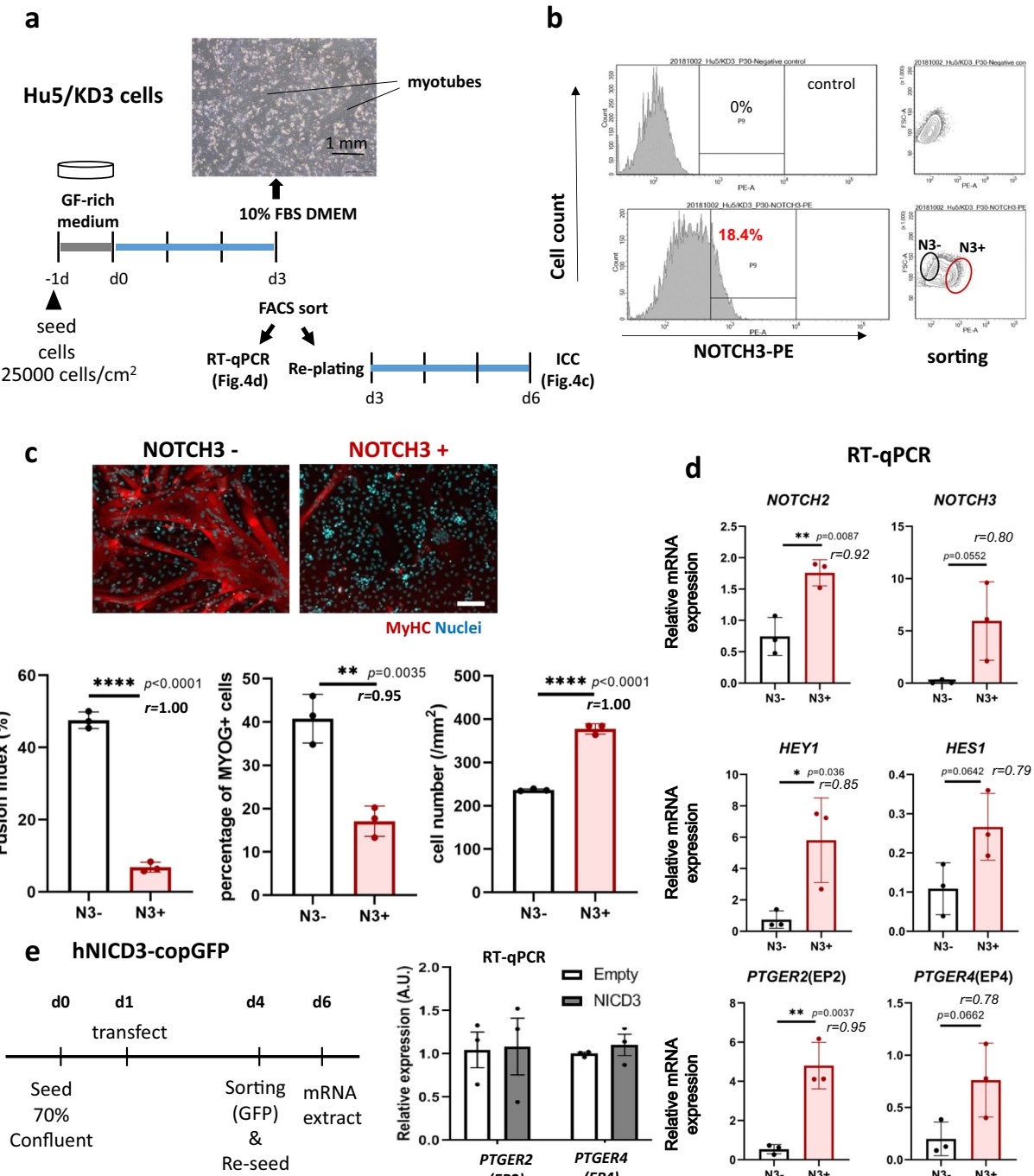

**Fig. 4 NOTCH3-high muscle progenitors were self-renewing, and highly expressed PTGER2 and PTGER4. a** Experimental design. Hu5/KD3 cells were seeded onto collagen type I-coated 60-mm dishes (25,000 cells/cm²) and cultured in 10% FBS/DMEM for 3 days. Then FACS sorting and qPCR analysis were performed. Sorted cells were replated and cultured for a further 3 days, fixed, and stained with MF20 (muscle myosin heavy chain) and DAPI (nuclei). **b** FACS plot. Sorted NOTCH3-high (N3+) and NOTCH-negative (N3−) fractions are shown by ellipses. **c** Representative muscle myosin heavy chain staining of differentiating NOTCH3-negative (N3−) and NOTCH3-positive (N3+) Hu5/KD3 cells. Cells were stained with MF20 (MyHC, red) and nuclei (DAPI, blue). Scale bar = 100 μm. Fusion index, percentage of MYOGENIN-positive cells, and cell numbers are shown as dot plots. Data are shown as mean ± S.D. and analyzed by unpaired two-tailed Student's *t*-test. *n* = 3 samples/group. Three views/sample were analyzed. Both *p* value and the effect sizes of Pearson's *r* correlation (*r*) are shown. **d** qPCR for *NOTCH2*, *NOTCH3*, *PTGER2*, *PTGER4*, *HEY1*, and *HES1* of NOTCH3-negative (N3−, blue ellipse in **b**) and NOTCH3-positive (N3+, red ellipse in **b**) cell fractions. Data are shown as mean ± S.D. and analyzed by unpaired two-tailed Student's *t*-test. *n* = 3 samples/group. Both *p*-value and the effect sizes of Pearson's *r* (*r*) are shown. **e** Hu5/KD3 cells were transfected with hNICD3-copGFP plasmid or a parental (empty) plasmid, and 3 days later, copGFP-positive cells were collected by FACS and re-seeded on collagen-coated dishes. Two days later, total RNA was extracted from the cells and qPCR for *PTGER2(EP2)* and *PTGER4(EP4)* was performed. Unpaired two-tailed Student's *t*-test, *n* = 3 samples/group.

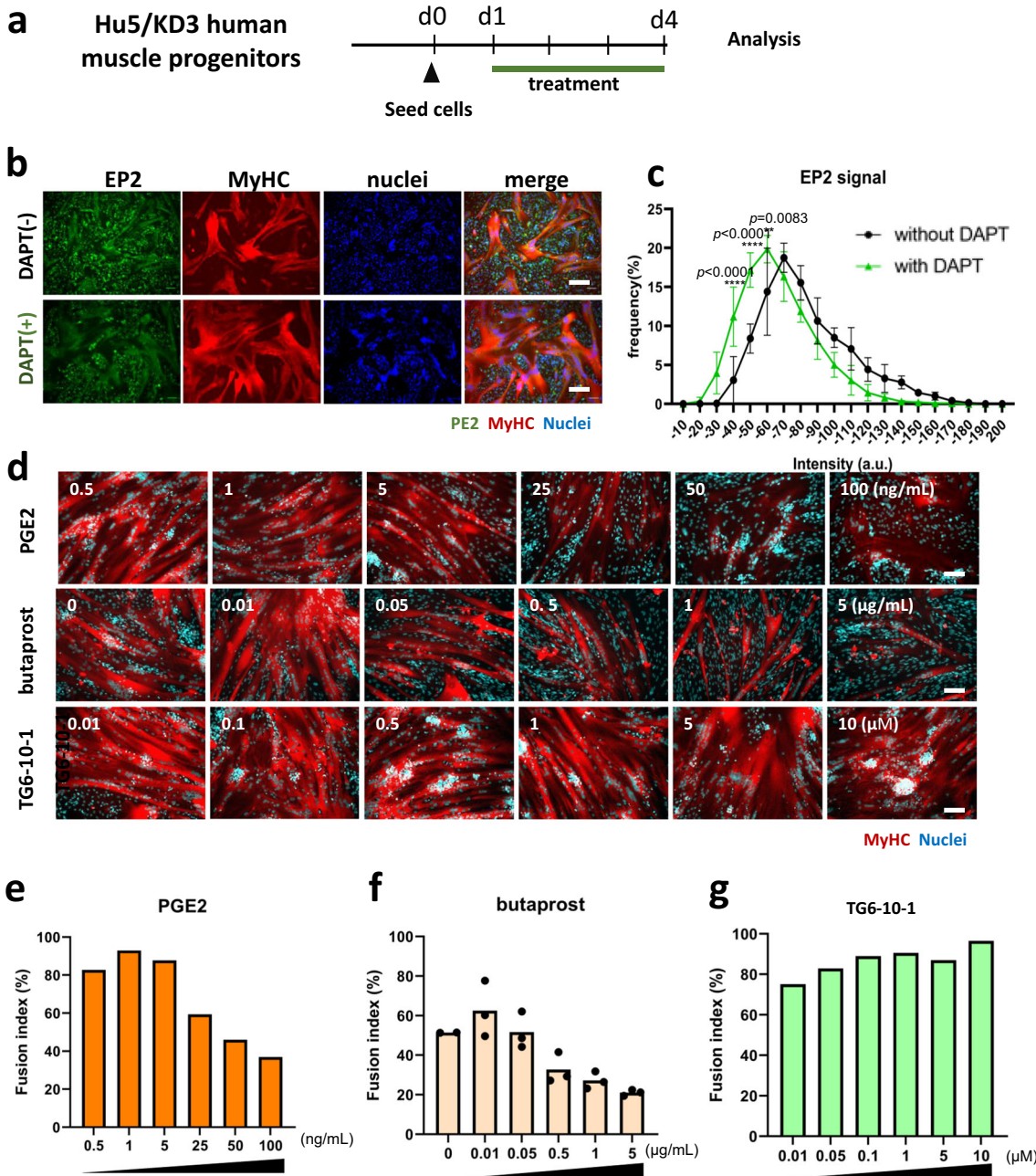

**Fig. 5 Effects of prostaglandin E2 (PGE2), EP2 agonist (butaprost), EP2 antagonist (TG6-10-1), and overexpression and knockdown of EP2 on myotube formation of a human myoblast cell line, Hu5/KD3 cells. a** Experimental design. Hu5/KD3 cells were seeded onto collagen type I-coated 12-well plates ($5 \times 10^5$ cells/well) and cultured in 10% FBS/DMEM medium with prostaglandin E2 (0.5–100 ng/ml), butaprost (EP2 agonist) (0.01–5 µg/ml), or PE2 antagonist, TG6-10-1 (0.01–10 µM) for 3 days. **b** Expressions of EP2 (green) and MyHC (MF20, red) on differentiating Hu/KD3 cells w/o DAPT treatment. Scale bar, 100 µm. **c** Histogram of EP2 signal of mononuclear cells treated with (green) or without (black) DAPT. Y-axis indicates the frequency. Data are shown as mean ± S.D. and analyzed by two-way ANOVA followed by Sidak's multiple comparisons test. More than 600 cells in each group were measured ($n = 3$ samples/group). **d** Representative MyHC staining of differentiating Hu5/KD3 cells treated with PGE2, butaprost, or TG6-10-1 at indicated concentrations. Scale bar, 100 µm. **e** Fusion index of Hu5/KD3 cells after treatment with PGE2. **f** Fusion index of Hu5/KD3 cells after treatment with butaprost. **g** Fusion index of Hu5/KD3 cells after treatment with TG6-10-1.

cells and Hu5/KD3 cells, and their concentration was not significantly changed by DAPT treatment (Supplementary Fig. 7). The expression level of COX-2 mRNA in Hu5/KD3 cells was also extremely low when examined by RT-qPCR (Supplementary Fig. 7d). Interestingly, PGE2 levels rapidly increased in the later stage of differentiation (Supplementary Fig. 7e). These results and EP2 overexpression experiments (Fig. 6a–d, Supplementary Fig. 5d–f) suggest that the PGE2-EP2 receptor signaling is mainly

regulated by the expression levels of the receptor rather than the PGE2 concentration in the early phase of differentiation. To understand the roles for PGE2 in the later stage of muscle differentiation, which is likely produced by multinucleated myotubes, further analysis needs to be performed.

**cAMP downstream of EP2 signaling did not promote self-renewal of muscle progenitors.** To clarify the downstream

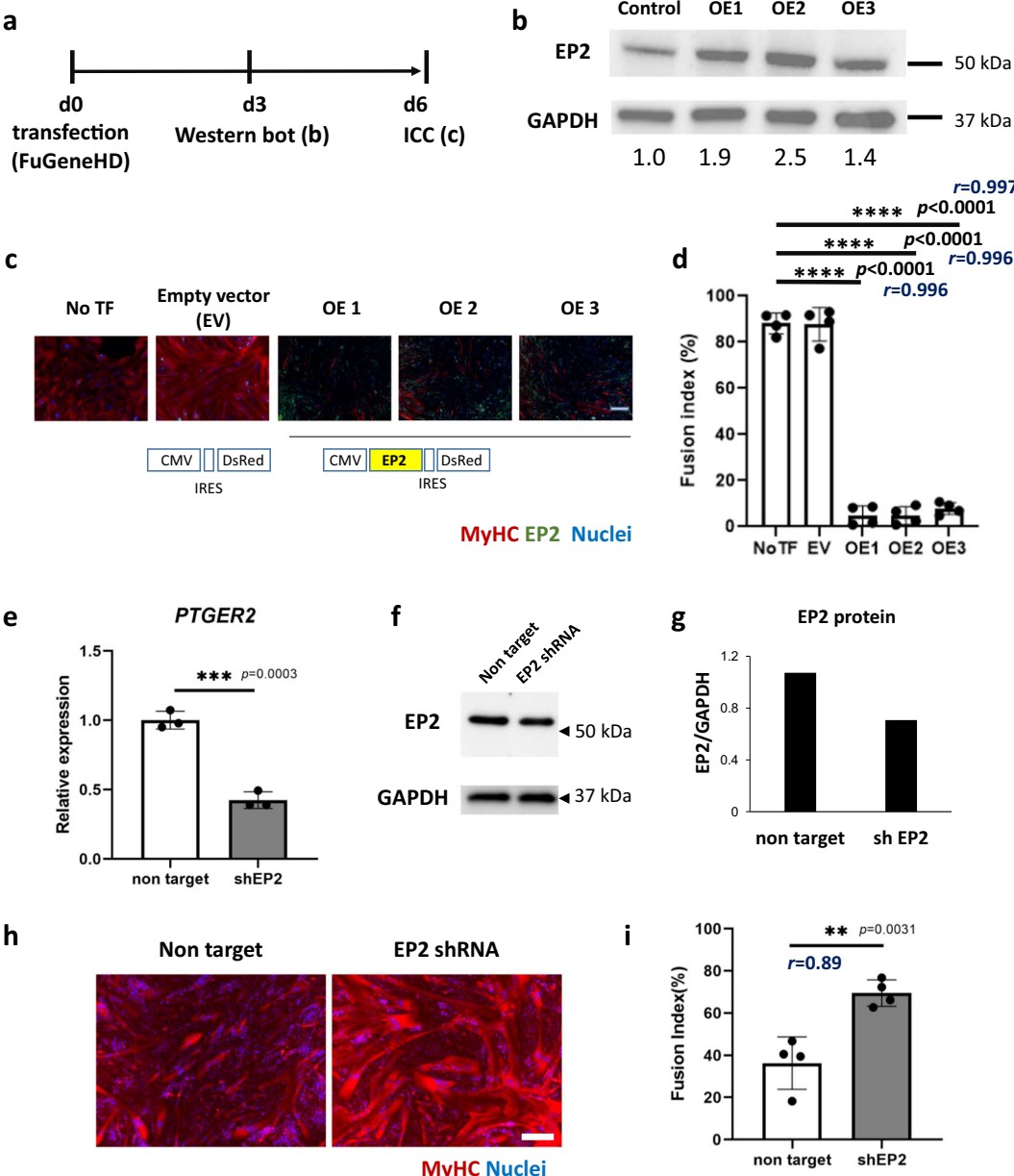

**Fig. 6 Overexpression and knockdown of EP2 in Hu5/KD3 cells. a** Experimental design of overexpression of EP2 in Hu5/KD3 cells. **b** Western blotting for EP2 (overexpression: OE1-3) and GAPDH (loading control) 3 days after transfection of the control vector (CMV-DsRed) or CMV-EP2-ires DsRed vectors. Relative expression levels of EP2 (after normalization to GAPDH) are shown below. **c** Representative image of myotubes (MF20, red) formed by Hu5/KD3 cells after transfection of the control vector (CMV-DsRed) or CMV-EP2-ires DsRed vectors. Scale bar, 100 μm. **d** Fusion index of (**c**). $n = 4$ samples/group. Dunnett's multiple comparisons test. **e** RT-qPCR for EP2 (PTGER2) of nontarget (control) short-hairpin (sh) RNA plasmid-transfected and EP2 shRNA plasmid-transfected Hu5/KD3 cells. Plasmids were introduced into Hu5/KD3 cells by electroporation. Data are shown as mean ± S.D. and analyzed by unpaired two-tailed Student's t-test, $n = 3$ samples/group. **f** Immunoblot analysis for EP2 and GAPDH in nontarget and shEP2 plasmid-transfected Hu5/KD3 cells. **g** EP2 signals were normalized to GAPDH. **h** Representative immunostaining of myotubes formed by nontarget shRNA plasmid or EP2 shRNA plasmid-transfected Hu5/KD3 cells with anti-myosin heavy chain antibody, MF20 (red) and nuclei (DAPI, blue). Scale bar = 200 μm. **i** Fusion index of Hu5/KD3 cells four days after transfection of shRNA plasmids. Data are shown as mean ± S.D. and analyzed by unpaired two-tailed Student's t-test, $n = 4$ independent experiments. Each dot is an average of 3 views/sample. Both p-value and the effect sizes of Pearson's r (r) are shown. **j** Full, uncropped images of western blots in **b** and **f** are shown in Supplementary Fig. 10.

signaling of EP2 receptor, we examined the effects of forskolin (adenylyl cyclase activator), cell membrane-permeable dibutyryl cyclic AMP (dbcAMP), which mimics endogenous cyclic adenosine monophosphate (cAMP), H-89, a PKA inhibitor, and ESI-09 (EPAC inhibitor), on the differentiation of Hu5/KD3 myogenic cells and hiPSC-derived myogenic cells. Unexpectedly, forskolin did not suppress the differentiation of muscle progenitors (Fig. 8e). Contrarily, dbcAMP stimulated the

differentiation of muscle progenitors (Fig. 8f, Supplementary Fig. 6g). H-89 and ESI-09 did not improve the differentiation of muscle progenitors (Fig. 8g, h, Supplementary Fig. 6h). 8-CPT-2Me-cAMP (selective activator of Epac) and 8-bromo-cAMP (selective activator of protein kinase A) had no significant effects on differentiation of myogenic cells (Supplementary Fig. 8a, b). As expected, PGE2, butaprost, and CAY10684 (EP4 agonist) all increased the intracellular cAMP level in Hu5/KD3 cells, but

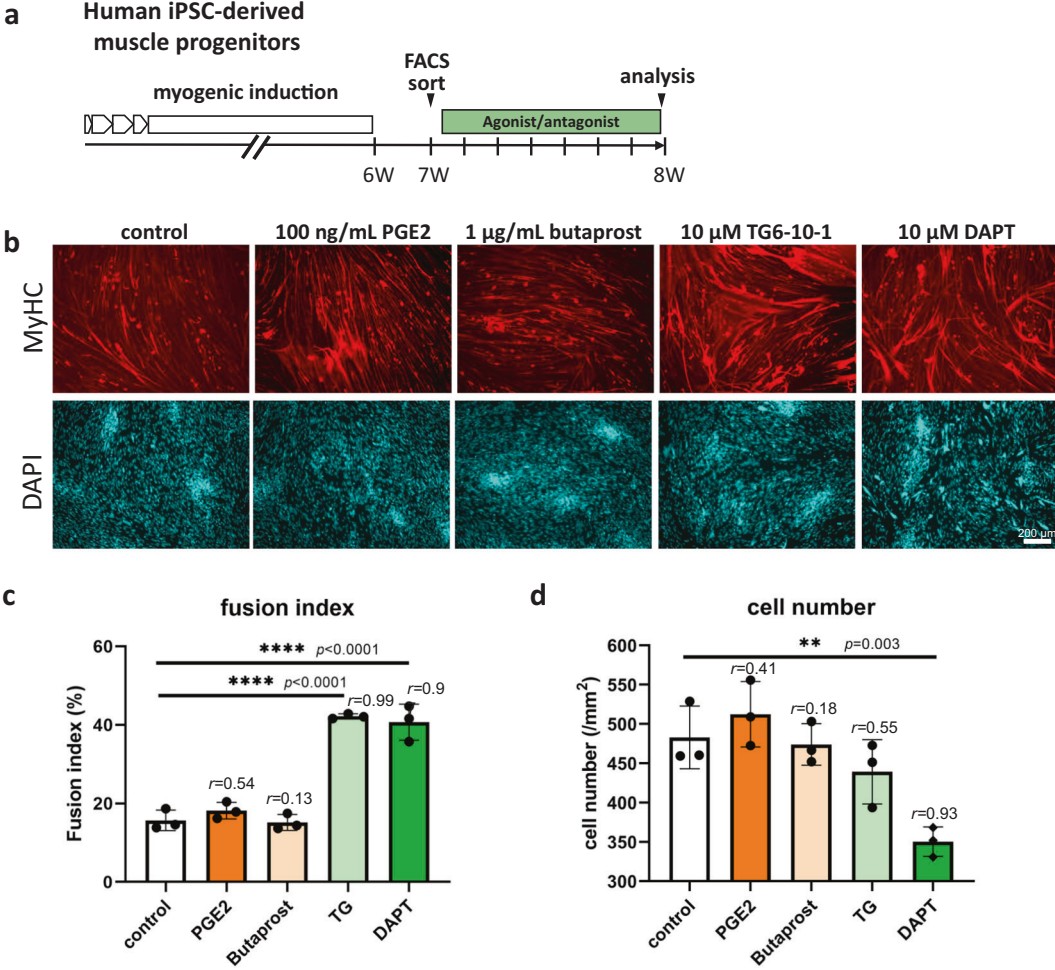

**Fig. 7 Blockage of EP2 signaling promoted differentiation of hiPSC-derived muscle progenitors. a** Experimental design. After 6 weeks myogenic induction, muscle progenitors were FACS-sorted and treated with EP2 agonists or an antagonist. **b** Representative images of myotubes formed by hiPSC-derived muscle progenitors treated with PGE2 (100 ng/ml), butaprost (1 μg/ml), TG6-10-1 (TG, 10 μM), and DAPT (10 μM). Cells fixed with 4% PFA were stained with MF20 (muscle myosin heavy chain, red) and nuclei (DAPI, blue). Scale bar, 200 μm. **c, d** Fusion index (%) (**c**) and cell numbers (/mm$^2$) (**d**). $n$ = 3 samples/group. Three views/sample were analyzed. Data are shown as mean ± S.D. and compared with controls by one-way ANOVA followed by Dunnett's multiple comparisons test.

CAY10684-treated myogenic cells fused well to form multinucleated myotubes (Fig. 8i–l). The results suggest that activation of the cAMP-PKA pathway does not promote self-renewal of myogenic progenitors.

## Discussion

DAPT improved myotube formation by a human muscle progenitor cell line, Hu5/KD3 cells, human primary myoblasts, and hiPSC-derived muscle progenitors. Gene expression analysis of DAPT-treated cells revealed that DAPT treatment downregulated *NOTCH3* and Notch effector genes, *HES1, HEYL*, and *HEY1*, indicating that Notch signaling is highly active in human muscle progenitors and inhibits their differentiation (Fig. 3, Table 1, Supplementary Fig. 2). Importantly, Kitzman et al. previously reported that inhibition of Notch signaling induces myotube hypertrophy in primary human myoblasts by recruiting a subpopulation of reserve cells[24]. In the present study, we observed that DAPT not only promoted differentiation of myogenic progenitors, but also promoted the efficiency of cell transplantation in xenotransplantation experiments (Fig. 2). We speculate that Notch inhibition improved the efficiency of cell transplantation by augmentation of fusion between donor cells and host

myofibers, possibly at the expense of self-renewal of muscle progenitors. If DAPT treatment suppresses replenishment of the satellite cell pool, the beneficial effects of DAPT might be short-term. The long-term effects of DAPT in cell transplantation, for example, after repeated muscle injury remain to be shown.

We found that DAPT treatment drastically downregulated NOTCH3 expression in human muscle progenitors. Recently, Notch3 was reported to mediate self-renewal of mouse C2C12 cells and primary myoblasts during differentiation and prevent their progression into the cell cycle[16]. To confirm the role of NOTCH3 in differentiation of human muscle progenitors, we examined NOTCH3 expression on muscle progenitors by FACS. NOTCH3 was induced in a fraction of the cells only when the cells were cultured at a high cell density, but not at a low cell density (Supplementary Fig. 4). As predicted, NOTCH3-negative cells quickly and robustly formed myotubes. In contrast, most NOTCH3-high cells remained mononuclear (Fig. 4). Thus, our results support Low's report on NOTCH3 function: NOTCH3 suppresses differentiation of a fraction of muscle progenitors to spare the stem cell fraction when a majority differentiate. Low et al. identified DLL4 as a ligand of Notch3. Since DLL4 expression in differentiating human myoblasts/ myotubes

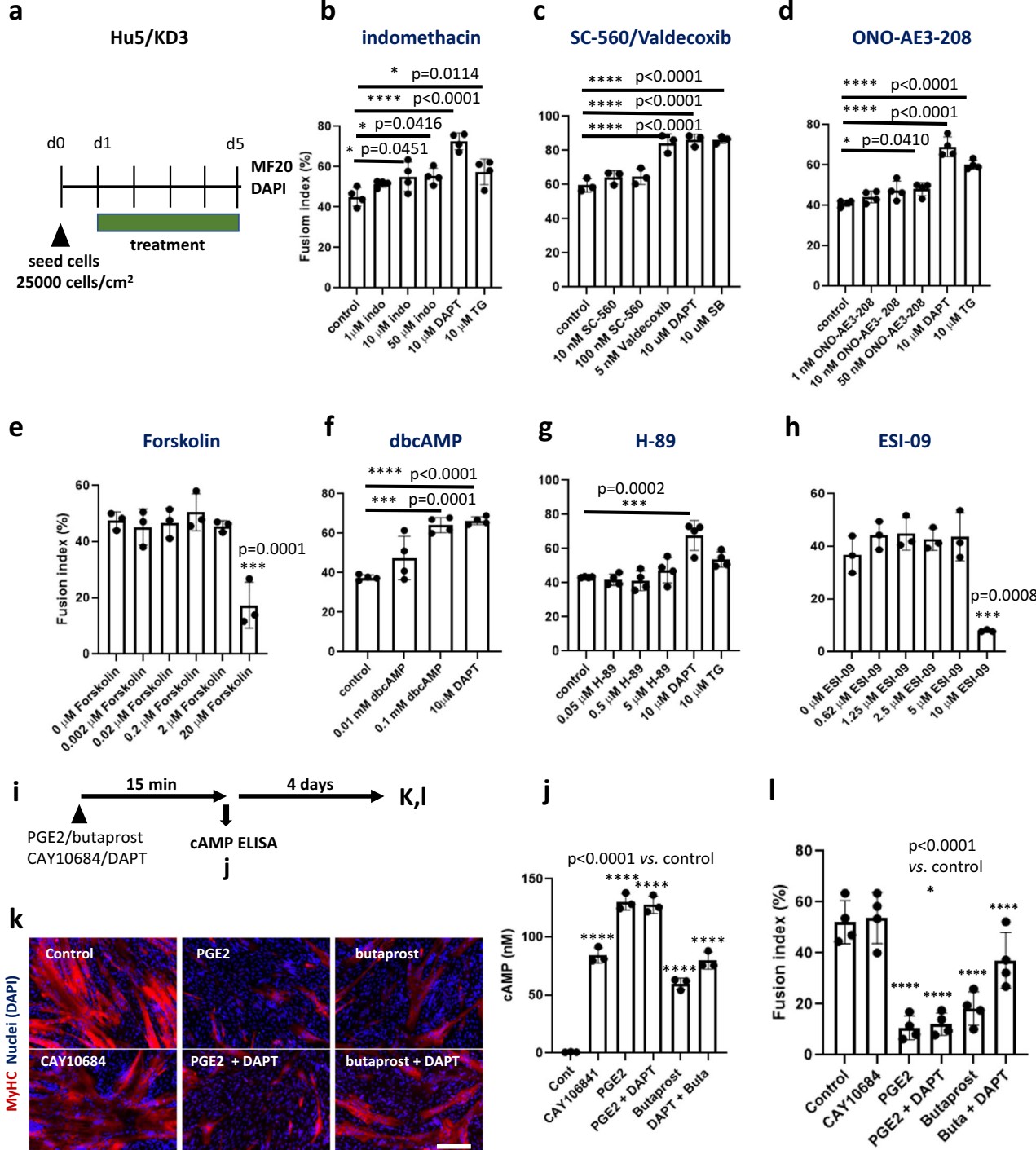

**Fig. 8 Signal transduction upstream and downstream of EP2 involved in self-renewal of muscle progenitors. a** Experimental design. Hu5/KD3 cells were seeded at $1 \times 10^5$ cells/well in 24-well plates, and treated with indicated reagents for 4 days. The cells were then immunostained with MF20 (muscle myosin heavy chain) and DAPI (nuclei). **b–h** Fusion index of Hu5/KD3 cells after 4 days treatment with indomethacin (**b**), Cox-1 inhibitor (SC-560), Cox-2 inhibitor (valdecoxib) (**c**), ONO-AE3-208 (EP4 antagonist) (**d**), forskolin (**e**), dbcAMP (**f**), H-89 (inhibitor of PKA) (**g**), or ESI-09 (Epac inhibitor) (**h**) are shown. Means ± SD. Dunnett's multiple comparisons test. $n = 3–4$ samples/group. Twenty micromolar of forskolin and 10 μM of ESI-09 had toxic effects on the cells: their morphology was quite different from usual unfused mononuclear myoblasts. **i** Experimental design. Fifteen minutes after addition of PGE2, butaprost, or CAY10684 with or without DAPT, the concentration of intracellular cAMP was measured by ELISA. Half of the cells were cultured for 4 days and examined by immunocytochemistry (**i**, **j**). **j** Intracellular cAMP levels 15 min after addition of PGE2 (25 ng/ml) with or without DAPT (1 μM), butaprost (500 ng/ml) with or without DAPT (1 μM), or CAY10684 (1 μg/ml, EP4 agonist). Means ± SD, triplicates. Tukey–Kramer test. $n = 3$ samples/ group. **k** Half of the cells shown in **i** were cultured for a further 4 days, fixed, and stained with MF20. Scale bar, 100 μm. **l** Fusion index of **i**. Note that CAY10684 upregulated cAMP at a similar level to butaprost, but treated cells fused well. Means ± SD. Tukey–Kramer test, $n = 4$ samples/group.

**Fig. 9 NOTCH3 and EP2 are key signaling molecules for self-renewal of muscle progenitors in differentiation phases.** In differentiation conditions, the majority of myogenic cells fuse to form multinucleated myotubes. NOTCH signaling upregulates the expression of EP2 and NOTCH3 in a fraction of muscle progenitors. Prostaglandin E2 activates EP2, and promotes self-renewal of muscle progenitors. The interplay between NOTCH3 signal and EP2 signals remains to be clarified. Fusako Sakai-Takemura et al. report that self-renewal of human muscle progenitors is regulated by NOTCH/EP2 signaling. The findings deepen our understanding on self-renewal of muscle stem cells and contribute to the development of treatments for Duchenne muscular dystrophy.

was almost undetectable in our RNA-seq analysis (Supplementary DATA), which Notch ligands serve as a ligand of NOTCH3 in human muscle progenitors remains to be shown. EP2 was exclusively expressed in NOTCH3-positive cells (Fig. 4d). Therefore, we examined whether NOTCH3-NICD directly upregulated the expression of the EP2 gene or not. Unexpectedly, overexpression of NOTCH3-NICD did not upregulate EP2 expression (Fig. 4e), suggesting that EP2 is upregulated by NOTCH members other than NOTCH3 in self-renewing cells. NOTCH2 is a candidate that upregulates the expression of the EP2 gene because NOTCH2 is highly expressed in NOTCH3-positive cells compared with NOTCH3-negative cells (Fig. 4d) and downregulated by DAPT treatment (Supplementary Fig. 9).

Prostaglandin E2 is a potent bioactive lipid messenger, which mediates diverse signals in physiological and pathological conditions. Recently, two groups reported that PGE2 activates proliferation of muscle myoblasts or muscle satellite cells via the EP4 receptor[25,26], but the roles of PGE2 in muscle differentiation are not well understood. Our results suggest that Notch signaling upregulated EP2 in a human muscle progenitor cell line, Hu5/KD3. A natural EP2 ligand, PGE2, and an EP2-specific agonist, butaprost, suppressed muscle differentiation. In contrast, a specific antagonist of EP2, TG6-10-1, improved myotube formation by Hu5/KD3 cells (Fig. 5d–g) and hiPSC-derived muscle progenitors (Fig. 7), suggesting that the NOTCH-EP2 axis regulates self-renewal of human muscle progenitors in differentiation-promoting conditions (summarized in Fig. 9).

Although our data suggest that the PGE2 produced by COX-2 regulated the cell fate decision, differentiation or self-renewal, the concentration of PGE2 in the culture medium was quite low and did not significantly change upon DAPT administration (Supplementary Fig. 7). Furthermore, the level of COX-2 mRNA was also extremely low and did not change upon DAPT administration (Supplementary Fig. 7). On the other hand, transient overexpression of EP2 by plasmid vectors mimicked the effects of butaprost or PGE2 (Fig. 6a–d, Supplementary Fig. 5d–f). In addition, downregulation of EP2 by shRNA plasmid vectors improved fusion of muscle progenitors (Fig. 6e–i). These data suggest that the EP2 signaling was regulated mainly at the expression level of EP2 receptors.

Both EP2 and EP4 activated adenyl cyclase and increased the intracellular cAMP concentration in the cells. However, an antagonist of EP4, ONO-AE3-208, did not improve the fusion index of Hu5/KD3 myogenic cells (Fig. 8d), suggesting that EP4 did not make a major contribution to self-renewal of myogenic cells.

Forskolin, an activator of adenylyl cyclase, did not inhibit muscle differentiation. H-89, a protein kinase A (PKA) inhibitor, or ESI-09 (Epac inhibitor) did not promote the fusion of myogenic cells, and dbcAMP promoted muscle differentiation of the cells (Fig. 8, Supplementary Fig. 6). 8-CPT-2Me-cAMP (selective activator of Epac) and 8-Bromo-cAMP (selective activator of protein kinase A) had no significant effects on differentiation of myogenic cells (Supplementary Fig. 8a, b). Together, these results suggest that EP2 promoted self-renewal of myogenic cells via cAMP-independent pathways.

In conclusion, we showed that Notch inhibition promoted differentiation of human muscle progenitors in vitro and in vivo. We also found an important role for Notch/PGE2/EP2 receptor signaling in regulation of the cell fate of myogenic progenitors. Molecular mechanisms by which EP2 promotes self-renewal of myogenic progenitors remain to be determined.

## Methods

**Ethical statement**. The research plans using human iPS cells and all experimental procedures using mice were approved by the Ethical Committees of the National Institute of Neuroscience, National Center of Neurology and Psychiatry (NCNP), Japan, and performed according to the guidelines.

**Cells and culture**. Hu5/KD3 cells are a human myoblast cell line, established using hTERT, mutant CDK4 (CDK4R24C), and cyclin D1[20]. The cells were maintained on collagen type I-coated dishes (Iwaki) in high glucose DMEM (Wako) supplemented with 20% fetal bovine serum (FBS, Gibco), 2% Ultroser G (Biosepta, Pall) as described[20]. For differentiation, cells were cultured in DMEM containing 10% FBS.

A human iPS cell line, 201B7, which was established from a healthy donor, was provided by S. Yamanaka at the Center for iPS Cell Research and Application (CiRA), Kyoto University[27]. hiPSCs were cultured with mitomycin-C-inactivated mouse embryonic fibroblasts (MEFs) on gelatin-coated dishes (Iwaki) in primate ES cell medium (ReproCELL) supplemented with 10 ng/ml FGF-2 (Pepro Tech). Prior to myogenic induction, hiPS cells were replated onto iMatrix-511 (Nippi)-coated 6-well plates in Stem Fit AK02N (Ajinomoto) supplemented with penicillin/streptomycin/amphotericin B (PSA, 1% v/v) (Wako).

Adult primary myoblasts (human skeletal muscle myoblasts) were purchased from Lonza (catalog #CC-2580) and cultured on collagen type I-coated dishes in 10% FBS/DMEM.

We routinely checked mycoplasma contamination by a PCR method (PCR Mycoplasma Detection Set cat#6601, Takara).

To inhibit Notch signal, cells were cultured in DMEM/10% FBS containing 10 μM DAPT (Sigma-Aldrich). Because DAPT was dissolved in DMSO (10 mM stock solution), 0.1% DMSO was added to the control culture. For inhibition of prostaglandin EP2 receptors, an EP2-specific antagonist, TG6-10-1 (Calbiochem), was added to the cell culture at 1–10 μM. To inhibit EP4 receptors, ONO-AE3-208 (1–50 nM; a gift from Ono Pharmaceutical Co. LTD.) was used. To stimulate EP2, prostaglandin E2 (Nacalai Tesque) (0.5–100 ng/ml) or an EP2 agonist, butaprost (Cayman) (0.01–5 μg/ml) was added to the culture. To stimulate EP4, CAY10684 (Cayman) was added to the culture. The medium was changed every day. For inhibition of PGE2 production, SC-560 (Cayman), valdecoxib (Tokyo Chemical Industry Co. Ltd. (TCI)), or indomethacin (Wako) was added to the culture. For inhibition of protein kinase A, H-89 (Cayman) was added to the culture. To examine the effects of cAMP on muscle differentiation, dbcAMP (Fujifilm) was used. Forskolin (TCI) was used to activate adenylyl cyclase. ESI-09 (Cayman) was

used for inhibition of Epac. 8CPT-2Me-cAMP (TOCRIS) and 8-bromo-cAMP (TOCRIS) were used to activate Epac and PKA, respectively.

**Derivation of muscle progenitors from hiPS cells**. Muscle progenitors were induced from human iPS cells as described[10]. In brief, after induction of the paraxial mesoderm[7], cells were cultured as floating spheres in Stemline for neural stem cells (Sigma-Aldrich, S3194) supplemented with 100 ng/ml FGF-2 (Pepro Tech), 100 ng/ml EGF (Pepro Tech), and 5 µg/ml heparin sodium[6]. Spheres were cut into 200-µm cubes by a McIlwain tissue chopper once a week (Mickle Laboratory Engineering). After 6-week induction, spheres were plated onto collagen type I-coated 10-cm plates (Iwaki) and cultured in 10%FBS/DMEM for 1 week.

**Immunocytochemistry (ICC)**. ICC was performed as described previously[10]. After fixation in 4% paraformaldehyde, cells were permeated with 0.1% TritonX-100 for 10 min at RT. After blocking with 5% goat serum (Cedarlane)/2% bovine serum albumin (BSA, Sigma-Aldrich) in PBS(-), cells were incubated with primary antibodies (100–500 dilution): muscle myosin heavy chain (MF20, R&D Systems), myogenin (Santa Cruz Biotechnology, FD5 or rabbit polyclonal), Pax7 (Santa Cruz Biotechnology, PAX7), MyoD (5.8 A or rabbit polyclonal; Santa Cruz Biotechnology), or anti-EP2 antibody (Abcam, rabbit monoclonal, EPR8030(B)) overnight at 4 °C. The next day, the cells were washed in PBS(-) and incubated with secondary antibodies (1000–2000 dilution): Alexa 568 goat-anti-mouse IgG2b or Alexa488 goat anti-rabbit IgG (Molecular Probes), for 2 h. Nuclei were stained with DAPI (Tokyo Chemical Industry). Images were taken with a KEYENCE BZ-9000 microscope and analyzed by BZ-II image analysis software and hybrid cell counting software (Keyence Corp.) or Image J (NIH).

**RNA isolation, cDNA synthesis, qPCR**. Total RNA was extracted from cells with Trizol (Invitrogen) or an RNeasy Mini Kit (Qiagen), reverse-transcribed into cDNA using a PrimeScript RT reagent kit (Perfect Real Time, Takara), and amplified by SYBR Premix EX Taq II (Til RNaseH Plus, Takara) and primer sets as shown in supplementary Table 1. The primers were designed to amplify a single band. The signals were recorded, and ΔCt (1/2^ (Cq of the housekeeping gene (GUSB) Cq)) was determined by a CFX Connect system (Bio-Rad). GUSB transcripts were used for normalization.

**RNA sequencing**. RNA sequencing was performed by Takara Bio (Lot ID: PR0938). The library was prepared by using TruSeq Stranded mRNA Library Prep Kit (Ilumina). Sequencing was done by using Ilumina HiSeq 2500 (100 bp, paired-end). Read alignment was performed by STAR (ver. 2.5.2b). Data analysis was performed by using Genedata Profiler Genome (ver. 10.1.15a). The gene expression level was normalized by FPKM (fragments per kilobase of exon per million reads mapped). Data were presented as log FPKM. RNA-seq data have been deposited in the ArrayExpress database at EMBL-EBI (www.ebi.ac.uk/arrayexpress) under accession number E-MTAB-8825.

**FACS analysis and cell sorting**. Cells were detached from the culture plates by treatment with 0.05% trypsin-1% EDTA (Gibco) and collected by a cell scraper (Iwaki), centrifuged, resuspended in wash buffer (2%FBS in PBS), filtered using 100-µm mesh (Falcon), and counted by a Countess cell counter (ThermoFisher). Cells ($0.1–1.0 \times 10^7$) were incubated with antibodies (1:200 dilutions) in 700 µl PBS containing 2% FBS (wash buffer) for 30 min at 4 °C, washed in wash buffer, incubated with propidium iodide briefly to stain dead cells, and analyzed and sorted using a BD FACSAria™ Fusion equipped with three lasers (405, 561, 640 nm) using FACS Diva (BD Bioscience, v 8.0) and FlowJo (BD Biosciences). The following antibodies were used in this study: CD57(HNK-1)-PE (clone TB03, Miltenyi Biotec), ERBB3-APC (clone REA508, Miltenyi Biotec), CD271-BB515 (clone C40-1457, BD Pharmingen), and human NOTCH3-PE (clone: MHN3-21, BioLegend).

**Plasmid transfection**. hNICD3(3xFLAG)-pCDF1-MCS2-EF1-copGFP was a gift from Brenda Lilly (Addgene plasmid # 40640)[28]. The parental pCDF1-MCS2-EF1-copGFP (System Biosciences) was used as a control. Transfection of the plasmid was performed using FuGeneHD (Promega), and copGFP-positive and -negative cells were sorted by FACS for RNA extraction.

CMV-PTGER2-tGFP (OriGene, RG210883)[29], a control CMV-tGFP (OriGene) or CMV-ires-GFP (Clontech) was introduced into 80–90% confluent Hu5/KD3 cells using FuGeneHD. CMV-EP2-ires-DsRed was constructed by insertion of human PTGER2 cDNA into the EcoRI site of CMV-ires-DsRed (Clontech) using an In-Fusion HD cloning kit (Takara). The human EP2 shRNA plasmid (sc-40171, Lot# E0709) and control shRNA plasmid (sc-108060) were purchased from Santa Cruz Biotechnology. Hu5/KD3 cells were electroporated with 10 µg of shRNA plasmid per $1.0 \times 10^6$ cells using a CUY21 Pro-Vitro electroporator (poring pulse: pulse voltage, 150 V; pulse width, 10 ms; pulse number, 50 ms; NEPA GENE, Ichikawa, Japan).

**cAMP and PGE2 ELISA**. The cells as well as the culture medium were collected. After sonication and centrifugation, the concentrations of cAMP in the supernatant were quantified by a cyclic AMP ELISA kit and a prostaglandin E2 ELISA kit, respectively, according to the manufacturer's protocols (Cayman Chemical).

**Western blot analysis**. Total muscle protein was extracted by sample buffer containing 15% glycerol, 1 mM dithiothreitol, 2% SDS, 125 mM Tris-HCl, and protease inhibitors (Roche). The protein concentration was determined using Bio-Rad Protein Assay Dye Reagent Concentrate (Bio-Rad Laboratories, Inc., Hercules, CA) with bovine serum albumin as a standard. Fifteen micrograms of protein were separated on gradient SDS-polyacrylamide gels (4–20% tris-glycine gel) and electrically transferred to a polyvinylidene difluoride membrane (Millipore). The blot was incubated with primary antibodies: anti-EP2 antibody (Abcam, rabbit monoclonal, ab167171, 1:1000 dilution) and anti-GAPDH antibody (Santa Cruz, goat polyclonal, sc-20357, 1:1000) and ECL™ Anti-Rabbit IgG, Horseradish Peroxidase linked F(ab′)₂ fragment (GE Healthcare, NA9340V, 1:1000 dilution) and HRP-Rabbit Anti-Goat IgG(H + L) (Invitrogen, 611620, 1:1000 dilution) were employed as secondary antibodies. The signals were detected using the ECL Prime Western Blotting Detection Reagent (GE Healthcare) and a ChemiDoc MP Imaging System (Bio-Rad). Data were analyzed by using Image Lab 6.0 (Bio-Rad).

**Mice and cell transplantation**. $NSG\text{-}mdx^{4cv}$ is a severely immunodeficient dystrophin-deficient mouse[30] provided by M. Kyba of the University of Minnesota. NOD/Scid mice were purchased from Nihon Clea. All mice were maintained in the specific pathogen-free animal facility at the National Institute of Neuroscience in Japan. Five-month-old male $NSG\text{-}mdx^{4Cv}$ mice were used for cell transplantation. Twenty-four hours prior to transplantation, tibialis anterior (TA) muscles were injured by injection of 50 µl of 2% $BaCl_2$ (Sigma-Aldrich). FACS-sorted cells were resuspended in PBS(−) supplemented with 10% Matrigel (BD Bioscience), 100 ng/mL LIF (Prospec)[31], 100 ng/mL HGF (Peprotec), and green fluorescent beads (Life Technologies) with or without 10 µM DAPT (Wako) or TG6-10-1 (Merck Millipore). Sixty microliters of a cell suspension containing 100,000–1,000,000 cells was injected into TA muscles of $NSG\text{-}mdx^{4Cv}$ male mice using a 29G needle (Terumo) under general anesthesia. In some experiments, DAPT solution (10 µM, 30 µl) or TG6-10-1 (10 µM, 30 µl) was injected every 3 days (total four times) after cell transplantation. After 4 weeks, the mice were sacrificed and TA muscles were dissected, quickly frozen in isopentane cooled by liquid nitrogen, and sectioned transversely by a cryostat (8-µm thick).

**Immunohistochemistry**. Cross-sections of TA muscle were fixed in cold acetone for 10 min and air-dried. After blocking with 5% goat serum and 2% BSA in PBS (-), tissue sections were incubated with antibodies to human lamin A/C (mouse monoclonal, NCL-LAM-A/C, Leica), human spectrin (mouse monoclonal, NCL-SPEC1, Leica), and dystrophin (rabbit polyclonal, Abcam) (100–400X dilution) overnight at 4 °C. The next day, primary antibodies were washed out with PBS(-), then incubated with Alexa 488 goat-anti-mouse IgG2b, Alexa568 goat anti-rabbit IgG, or Alexa568-goat anti-mouse IgG2aκ (Molecular Probes) for 2 h, and mounted in Vecta Shield containing DAPI (Vector). Images were recorded with a KEYENCE BZ-9000 microscope and analyzed using hybrid cell count software (Keyence Corp.) or Image J.

**Statistics and reproducibility**. We did not exclude any data from the analysis. Data were expressed as the mean ± standard deviation (SD), and analyzed and plotted using the GraphPad Prism 8 software. The significance of differences between two groups was analyzed by the unpaired two-tailed Student's t-test. Comparisons of multiple experimental groups were performed by two-way ANOVA followed by Sidak's multiple comparisons, or one-way ANOVA followed by Dunnett's or Tukey–Kramer's multiple comparisons test. $*p < 0.05$, $**p < 0.01$, $***p < 0.001$, and $****p < 0.0001$. Effect sizes were calculated using the following formula: $r = \sqrt{\frac{t^2}{t^2 + df}}$ (Pearson's r correlation). Raw data and other statistic parameters are presented in Supplementary DATA.

**Reporting summary**. Further information on research design is available in the Nature Research Reporting Summary linked to this article.

## Data availability

All the data of this study are shown in the main text and supplementary information files. The RNA-seq data were deposited in ArrayExpress with accession No. E-MTAB-8825. The source data underlying the main figures are presented in Supplementary Data. Any additional source data or material used in this study can be obtained from the corresponding author upon reasonable request.

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

## Acknowledgements
We thank Dr. Tachimori for advice on Statistics. This study is supported by (1) Research Funds for "Development of cell transplantation methods for refractory muscle diseases" (Projects for Technological Development) and "Research on refractory musculoskeletal diseases using disease-specific induced pluripotent stem (iPS) cells" from the Research Center Network for Realization of Regenerative Medicine, Japan Science and Technology Agency (JST), and Japan Agency for Medical Research and Development (AMED), (2) Grants-in-aid for Scientific Research (C) (24590497, 16K08725, 19K075190001) from the Ministry of Education, Culture, Sports, Science and Technology (MEXT), Japan and (3) Intramural Research Grants (27-7 and 30-9) for Neurological and Psychiatric Disorders of NCNP.

## Author contributions
F.S.T., K.N., and A.E. induced myogenic progenitors, performed FACS sorting, examined gene expression, and performed cell transplantation experiments. F.S.T., K.N., Y.M., and K.K. performed cell culture and immunohistochemistry, and analyzed the data. N.H. established Hu5/KD3 cells and optimized culture condition and gene transfer. F.S.T. and Y.M.S. designed the experiments, wrote the paper, and prepared the figures. Y.M.S. and S.T. supervised the whole research project.

## Competing interests
The authors declare no competing interests.
