## [Peer Review File · Communications Biology]

Reviewers' comments:

Reviewer #2 (Remarks to the Author):

This study examined the role of DAPT, which promotes myoblast fusion to promote engraftment of myoblasts and the manner by which DAPT promotes myoblast fusion. The authors conclude that DAPT enhances engraftment in a mouse model and acts by downregulating NOTCH3 and prostaglandin E2 receptor 2. The results are potentially novel, but further clarification is needed to support the claims.

The statistical analysis was appropriate although details are needed about the analysis of the RNASeq results.

The two goals of the paper detract from the overall impact of the study. One set of experiments examine engraftment of myoblasts after treatment with DAPT which is known to promote myoblast fusion. The second goal, and the title of the paper, examined signaling pathways and provides evidence that DAPT acts by blocking NOTCH3 and prostaglandin EP2 signaling. While there is some relationship between the two objectives, there are a number of important issues to address with respect to engraftment that may or may not be relevant to a study of signaling pathways. These issues are:

1. The choice of 10 μ M DAPT caused aberrant fusion of myoblasts as shown by the large number of nuclei and larger myofiber size in Figures 1e and 1I with hiPSCs and a human myoblast line. Fusion looks somewhat better with primary human myoblasts treated with 10 μ M DAPT (Supplementary Figure 1b), although there is still significant variability in myotube diameter.
2. For engraftment studies, important to know whether the increased engraftment due to addition of DAPT leads to myofiber maturation and functional improvements in the muscle.
3. Ideally a dose-response study should be performed to identify conditions that improve engraftment and function.

Thus, studies are inconclusive as to whether the use of DAPT is suitable to enhance engraftment for therapeutic applications.

The results with respect to the role of NOTCH3 and PGE2 receptor 2 are suggestive but additional support is needed. For the experiments in Figure 4, it was unclear whether the FACS and RT-PCR were performed after sorting and at the time of fusion to ensure that recovery of NOTCH 3 did not occur. Further, NOTCH2 levels had declined in the Notch3 negative cells, which could affect fusion.

The studies on the role of PGE2 need further clarification. There are reports the PGE2 promotes proliferation of myoblasts but Butaprost had no effect and EP4 seems to be responsible, (e.g. Cell Cycle 14(10): 1507-1516 (2015)). Also, the dose-response of PGE2 on the fusion index is very different than that observed by Beaulieu et al. (2012. Neurobiology of Diseases 45 (1) 122-129), Figure 3A, which showed a 50% inhibition of human wild type myoblast fusion at 0.5 ng/mL PGE2. The response of the hiPS-derived muscle progenitors differs from the results with primary cells (Figure 6c). Measurement of the PGE2 receptors and knockdown of specific PGE2 receptors may help to clarify the role of PGE2 in proliferation

Specific Comments

1. Figure 1d. Clarify when the fusion index was measured.
2. Figure 5e-5g. Provide an explanation as to why the fusion index for low concentrations of drug or no drug is high relative to results reported in Figure 1d for the Hu5/KF3 cells.
3. For the RNA-Seq analysis, how well does the Hu5/KD3 human myoblast line compare with primary human myoblasts? The response to PGE2 and Butaprost may go through a maximum and differ from that of TG6-10-1 but the high baseline may make it difficult to discern an effect due to the drugs on

myoblast fusion.

4. Additional details are needed about RNA Sequencing including method of library preparation, read alignment, software used and FDR levels used.

Reviewer #3 (Remarks to the Author):

Summary:

The manuscript entitled "Prostaglandin EP2 receptor downstream of NOTCH signaling inhibits differentiation of human skeletal muscle progenitors in differentiation condition" by Sakai-Takemura et al. shows that PGE2-EP2 signaling promotes self-renewal of human muscle progenitor cells downstream of NOTCH. They showed that the inhibition of EP2 receptor activation as well as NOTCH signaling facilitated differentiation of muscle progenitors with cell fusion as an index. However, there is insufficient evidence, the logic is only superficial, and manuscript is not clearly written as pointed out below. To this reviewer, this manuscript remains too preliminary, and therefore not suitable for publication in the Communications Biology.

Major concerns:

1) Lack of results regarding the PG synthesis pathway:

The authors concluded that NOTCH signaling suppresses differentiation of muscle progenitor cells, and that such effect is mediated by PGE2-EP2 receptor signaling. However, the authors did not observe the PGE2 production and its synthetic pathway. What cell type (muscle progenitor or neighboring cell) produces PGE2?

In these cells, is PGE2 release as well as cyclooxygenase (COX; COX-1 or COX2) gene expression increased upon some stimuli or DAPT treatment?

In any case, the treatment of cell with an aspirin-like drug such as indomethacin should mimic the effect of DAPT. How about COX-1-selective and COX-2-selective inhibitor?

In addition, the authors should show the significant amount of PGE2 is really produced in the culture media at the corresponding levels showing pharmacologically inhibitory effect on fusion index (25 ng/mL, approximately 70 nM) as shown in Fig. 5e.

2) Only EP2 but not EP4 is really involved?

Why the effective doses were different between PGE2 and butaprost? To this reviewer, PGE2 appeared to be more effective than butaprost. Since the authors detected the expression of EP4, another cAMP-coupled PGE receptor, in NOTCH3-positive cells, the authors should evaluate potential contribution of EP4 receptor signaling to PGE2-induced effect. Moreover, the authors should show whether EP2 receptor is functionally expressed in muscle progenitor cells: PGE2 as well as butaprost (and also an EP4 agonist) is able to increase cAMP levels? Furthermore, such a cAMP-producing activities of PGE2 and butaprost were not detected in DAPT-treated cells?

3) Lack of verification of cAMP and downstream pathways:

The authors discussed EP2-derived cAMP may participate in self-renewal of muscle progenitor cells (page 5, line 3 from bottom). If so, the effect of PGE2 or butaprost should be mimicked by membrane-permeable cAMP such as dibutyryl cAMP (dbcAMP). Conversely, the effects of PGE2 and butaprost are inhibited by H-89, an PKA inhibitor?

The effect of DAPT is reversed by PGE2, butaprost, an EP4 agonist, and dbcAMP? Furthermore, even in the absence of NOTCH3, is dbcAMP able to maintain the self-renewal state in the muscle progenitors?

Minor points:

1. Through the manuscript, some parts of text are written in red, what does it mean?
2. Page 2, line 2, 'tert. butyl' should be 'tertial butyl'.
3. Page 2, line 6, 'prostaglandin' should be 'prostaglandin (PG)'.
4. Page 5. Line 5, Fig. 2d should be Fig. 3d.
5. Page 6, line 13, the authors should cite reference of an EP2-antagonist, TG6-10-1 (Jiang, J., et al. PNAS 110: 3591, 2013).

Answers to the reviewers' comments:

We deeply appreciate the reviewers' valuable comments on our manuscript. According to the comments, we performed additional experiments and obtained several new findings, which further support our hypothesis. The revised parts are in red in the text. This is a list of the figures we revised or newly added to the revised manuscript.

Revised or new figures		
Fig. 4a, e	new data (4e) were added	Overexpression of NICD3 and the effects on expression of EP2
Fig. 7	new	(i) Effects of COX-inhibitors, dbcAMP, H-89 (PKA inhibitor), and an EP4 antagonist (ONO-AE3-208) on differentiation of human muscle progenitors (Hu5/KD3 cells) (ii) cAMP production in cells treated with PGE2, butaprost, or CAY10684 and muscle differentiation of the treated cells
Fig. 8	new	Treatment of Hu5/KD3 cells with A419259 or SB203580 with or without butaprost
Fig. 9	revised	Graphic summary
Supplementary Fig. 3 (related to Fig. 3)	new	RT-qPCR of 10 genes, which were differently expressed in DAPT-treated and untreated Hu5/KD3 cells, of DAPT-treated or untreated human primary myoblasts
Supplementary Fig. 6 (related to Fig. 7)	new	Effects of COX-inhibitors, dbcAMP, H-89 on differentiation of human iPSC-derived muscle progenitors
Supplementary Fig. 7	new	RT-qPCR analysis for NOTCH1 , 2 , 3 , EP2 , and EP4 in myoblasts and myotubes
Supplementary Fig. 8	new	ELISA of PGE2 in the culture medium of the cells treated with DAPT- or COX-inhibitors & RT-PCR analysis for COX-2 mRNA in Hu5/KD3 cells treated with PGE2 or DAPT
Supplementary Fig. 9 (related to Fig. 7)	new	ELISA of cAMP in Hu5/KD3 cells treated with PGE2, CAY10684, or butaprost for three days.
Supplementary	new	Effects of inhibitors of Src, PI3K, JNK, GSK-3 β , EGFR,

Fig. 10 (related to Fig. 8)		or p38 MAPK on differentiation of Hu5/KD3 cells
--	---

The following are answers to the reviewer's comments.

Reviewers' comments:

Reviewer #2 (Remarks to the Author):

This study examined the role of DAPT, which promotes myoblast fusion to promote engraftment of myoblasts and the manner by which DAPT promotes myoblast fusion. The authors conclude that DAPT enhances engraftment in a mouse model and acts by downregulating NOTCH3 and prostaglandin E2 receptor 2. The results are potentially novel, but further clarification is needed to support the claims.

The statistical analysis was appropriate although details are needed about the analysis of the RNASeq results.

We added details about the analysis of the RNASeq results to the revised Ms.

The two goals of the paper detract from the overall impact of the study. One set of experiments examine engraftment of myoblasts after treatment with DAPT which is known to promote myoblast fusion. The second goal, and the title of the paper, examined signaling pathways and provides evidence that DAPT acts by blocking NOTCH3 and prostaglandin EP2 signaling. While there is some relationship between the two objectives, there are a number of important issues to address with respect to engraftment that may or may not be relevant to a study of signaling pathways.

These issues are:

- 1. The choice of 10 μ M DAPT caused aberrant fusion of myoblasts as shown by the large number of nuclei and larger myofiber size in Figures 1e and 1l with hiPSCs and a human myoblast line. Fusion looks somewhat better with primary human myoblasts treated with 10 μ M DAPT (Supplementary Figure 1b), although there is still significant variability in myotube diameter.**
- 2. For engraftment studies, important to know whether the increased engraftment due to addition of DAPT leads to myofiber maturation and functional improvements in the muscle.**
- 3. Ideally a dose-response study should be performed to identify conditions that improve engraftment and function. Thus, studies are inconclusive as to whether the use of DAPT is suitable to enhance engraftment for therapeutic applications.**

As the reviewer pointed out, the results in the original manuscript would not convince the readers of the usefulness of DAPT in cell transplantation because the percentage of dystrophin-positive myofibers was still low. In the revised manuscript, we focused on the clarification of the NOTCH-EP2 signaling pathway in myogenic cells.

The results with respect to the role of NOTCH3 and PGE2 receptor 2 are suggestive but additional support is needed. For the experiments in Figure 4, it was unclear whether the FACS and RT-PCR were performed after sorting and at the time of fusion to ensure that recovery of NOTCH 3 did not occur. Further, NOTCH2 levels had declined in the Notch3 negative cells, which could affect fusion.

RT-qPCR was performed on mRNAs extracted from freshly sorted NOTCH3-positive (N3+) and NOTCH3-negative cells (N3-). The fusion index was evaluated 3 days after re-plating sorted cells. We revised the experimental design shown in **Fig. 4a**.

To clarify the relationship between NOTCH3 and EP2, we overexpressed the intracellular domain of NOTCH3 (NICD3) in Hu5/KD3 cells. NICD3 did not increase EP2 expression, suggesting that NOTCH3 is not upstream of EP2. NOTCH2 is a candidate for upregulation of EP2, because NOTCH2 is expressed at a much higher level in myoblasts than in myotubes (**our supplementary Fig. 7** and Kitzmann et al., 2006), and, as the reviewer pointed out, NOTCH2 expression was higher in NOTCH3-positive cells than in NOTCH3-negative cells (**Fig.4d**). Our previous model of the NOTCH3-EP2 axis was revised (**new Fig. 9**).

The studies on the role of PGE2 need further clarification. There are reports the PGE2 promotes proliferation of myoblasts but Butaprost had no effect and EP4 seems to be responsible, (e.g. Cell Cycle 14(10): 1507-1516 (2015)).

Another EP2 agonist, ONO-AE1-259-1 did not stimulate the proliferation of hiPSC-derived myogenic cells and human primary myoblasts in growth condition (data not shown). The results are consistent with previous reports using mouse primary myoblasts (e.g. *Cell Cycle* 14: 1507-1516, 2015). Butaprost did not increase the numbers of myogenic cells in the differentiation conditions (**Fig. 6**). Thus, EP2 does not stimulate proliferation of the myogenic cells in either proliferation or differentiation condition. We think that EP2 can promote self-renewal of muscle progenitors without promoting cell proliferation.

In this study, we added EP4 agonist. ONO-AE3-208, to the cell cultures at full confluency in order to evaluate the effects on differentiation of muscle progenitors but not on

proliferation. ONO-AE3-208 did not promote muscle differentiation of the cells (**Fig. 7**). The result suggests that the contribution of EP4 to decision of cell fate (differentiate or stay undifferentiated) of muscle progenitors in differentiation-promoting condition (i.e., full confluency and low levels of growth factors) is relatively small.

Also, the dose-response of PGE2 on the fusion index is very different than that observed by Beaulieu et al. (2012. *Neurobiology of Diseases* 45 (1) 122-129), Figure 3A, which showed a 50% inhibition of human wild type myoblast fusion at 0.5 ng/mL PGE2.

Beaulieu *et al.* showed that 10–100 nM PGE2 (3.5–35 ng/ml) suppressed fusion by nearly 50%. In our experiments, the 50% fusion index was achieved by 25 ng/ml of PGE2 (**Fig. 5**). We think the dose response to PGE2 in their paper is not different from ours.

The response of the hiPS-derived muscle progenitors differs from the results with primary cells (Figure 6c). Measurement of the PGE2 receptors and knockdown of specific PGE2 receptors may help to clarify the role of PGE2 in proliferation.

We appreciate the reviewer's comments. The much lower fusion index in hiPSC-derived muscle progenitors compared with Hu5/KD3 or primary myoblasts is probably due to strong inhibition of their muscle differentiation by TGF- β signaling, which can be reversed by TGF- β inhibitor, SB431542 (Hicks et al., *Nat Cell Biol.* 20:46-57, 2018; Sakai-Takemura et al., *Sci Rep.* 8: 6555.2018). This makes detailed analysis of the EP2 signaling pathway in hiPSC-derived muscle progenitors a little difficult. We think that EP2 signaling and TGF- β signaling have some common downstream targets in hiPSC-derived muscle progenitors, because SB431542 masked the effects of EP2 antagonists (TG6-10-1) (data not shown).

Instead of a knockdown of EP2, which did not work well in our hands, we are now generating conventional knockout mice in which the EP2 gene can be inactivated in PAX7-positive muscle progenitors and muscle satellite cells. We should be able to report more details on the EP2-mediated signaling that regulates the cell fates of muscle progenitors in the near future.

Specific Comments

1. Figure 1d. Clarify when the fusion index was measured.

The fusion index was measured at Day 10. We added this description to the figure legend to Fig. 1 of the revised manuscript.

2. Figure 5e-5g. Provide an explanation as to why the fusion index for low concentrations of drug or no drug is high relative to results reported in Figure 1d for the Hu5/KF3 cells.

The fusion index of Hu5/KD3 cells is greatly affected by many factors, such as cell densities, timing of changing the medium (from 20%FBS/2% Ultrosor G/DMEM to 10% FBS/DMEM), timing of addition of the agonists /antagonists, and timing of evaluation. In Figure 1d, DAPT was added to the culture when the cell density was lower than in Figure 5e-5g. In any case, improvement of muscle fusion by DAPT was constantly observed in our experiments.

3. For the RNA-Seq analysis, how well does the Hu5/KD3 human myoblast line compare with primary human myoblasts? The response to PGE2 and Butaprost may go through a maximum and differ from that of TG6-10-1 but the high baseline may make it difficult to discern an effect due to the drugs on myoblast fusion.

We showed the RT-qPCR results of primary human myoblasts (**supplementary Fig. 3**), confirming that the genes differently expressed in DAPT-treated Hu5/KD3 human myoblasts were also differently expressed in primary human myoblasts.

Indeed, the baseline of the fusion index of the Hu5/KD3 human myoblast line was high, but we successfully evaluated effects due to the drugs on myoblast fusion (**Fig. 7**). We also evaluated the effects of TG6-10-1 on hiPS cell-derived myogenic cells (**Fig. 6**). TG6-10-1 improved fusion of hiPSC-derived muscle progenitors. We think that the results support the findings using Hu5/KD3 cells.

4. Additional details are needed about RNA Sequencing including method of library preparation, read alignment, software used and FDR levels used.

We added details about RNA sequencing to the Methods of the revised manuscript.

Reviewer #3 (Remarks to the Author):

Summary:

The manuscript entitled “Prostaglandin EP2 receptor downstream of NOTCH signaling inhibits differentiation of human skeletal muscle progenitors in differentiation condition” by Sakai-Takemura et al. shows that PGE2-EP2 signaling promotes self-renewal of human muscle progenitor cells downstream of NOTCH. They showed that the inhibition of EP2 receptor activation as well as NOTCH signaling facilitated differentiation of muscle progenitors with cell fusion as an index. However, there is insufficient evidence, the logic is only superficial, and manuscript is not clearly written as pointed out below.

To this reviewer, this manuscript remains too preliminary, and therefore not suitable for publication in the Communications Biology.

Major concerns:

1) Lack of results regarding the PG synthesis pathway:

The authors concluded that NOTCH signaling suppresses differentiation of muscle progenitor cells, and that such effect is mediated by PGE2-EP2 receptor signaling.

However, the authors did not observe the PGE2 production and its synthetic pathway.

What cell type (muscle progenitor or neighboring cell) produces PGE2?

TG6-10-1 promoted the differentiation of a myogenic cell line, Hu5/KD3 cells, and FACS-sorted CD271+ERBB3+ muscle progenitors derived human iPS cells, suggesting that myogenic cells themselves produce PGE2. But in the muscle regeneration process of the mouse model, PGE2 levels peaked at 3 days after Notexin injection (Ho *et al.*, *PNAS* 114, 6675-6684, 2017), suggesting that non-myogenic cells such as mesenchymal progenitor cells and inflammatory cells could be a major source of PGE2.

In these cells, is PGE2 release as well as cyclooxygenase (COX; COX-1 or COX-2) gene expression increased upon some stimuli or DAPT treatment?

We measured PGE2 levels using a prostaglandin E2 ELISA kit (Cayman, Item No.514010). PGE2 levels were extremely low in the culture medium of Hu5/KD3 cells (13–17 pg/ml), or in the culture medium of differentiating hiPSC-derived muscle cells (8–15 pg/ml) (**supplementary Fig. 8**). Therefore, it was difficult to determine whether DAPT down-regulates or up-regulates PGE2 production. The expression levels of COX-1 and COX-2 mRNA were also extremely low when examined by RT-qPCR, and down- or up-regulation

by DAPT treatment was not evident in Hu5/KD3 cells (**supplementary Fig. 8**). These results are consistent with RNAseq data, in which the signal levels of COX-1 and COX-2 genes were low and not down-regulated or up-regulated with DAPT treatment (**data not shown**). We speculate that the PGE2-EP2 receptor signaling is mainly regulated at the expression levels of the receptors.

In any case, the treatment of cell with an aspirin-like drug such as indomethacin should mimic the effect of DAPT. How about COX-1-selective and COX-2-selective inhibitor?

Indomethacin improved the fusion of Hu5/KD3 cells and hiPSC-derived muscle progenitors at a level similar to TG6-10-1, suggesting that the EP2 receptor was stimulated with PGE2 produced by COX-1 and/or COX-2 (**Fig. 7**). COX-2-selective valdecoxib, but not COX-1-selective SC-560, promoted differentiation of Hu5/KD3 myogenic cells (**Fig. 7**) and hiPSC-derived myogenic progenitors (**supplementary Fig.6**), suggesting that COX-2 is mainly involved in NOTCH signal-mediated suppression of muscle differentiation.

In addition, the authors should show the significant amount of PGE2 is really produced in the culture media at the corresponding levels showing pharmacologically inhibitory effect on fusion index (25 ng/mL, approximately 70 nM) as shown in Fig. 5e.

We measured the concentration of PGE2 by ELISA. The level was low in the culture medium of hiPSCs-derived myogenic cells (8–15 pg/ml) or Hu5/KD3 cells (13–17 pg/ml) (**supplementary Fig.8**). We added much higher levels of PGE2 in our experiments, but experiments using specific inhibitors indicated that PGE2 activated the EP2 receptor to promote self-renewal of muscle progenitors.

2) Only EP2 but not EP4 is really involved?

Why the effective doses were different between PGE2 and butaprost? To this reviewer, PGE2 appeared to be more effective than butaprost.

The butaprost we used (Cayman, item No.13740) is reported to bind with about 1/10 the affinity of PGE2 to the recombinant murine EP2 receptor (Br. J. Pharmacol, 122, 217-224, 1997). In COS cells expressing the human EP2 receptor, the EC₅₀ for the stimulation of cAMP by butaprost is about 5 μM, while the EC₅₀ for PGE2 is 43 nM (Mol. Pharmacol. 46, 213-220, 1994). Butaprost was much less effective than PGE2. On the other hand, ONO-AE1-259-1, another EP2 agonist, severely inhibited muscle differentiation at a much lower concentration (the fusion index was less than 10% in the presence of less 0.2 nM, data not shown), suggesting the importance of EP2 in the regulation of muscle differentiation.

Since the authors detected the expression of EP4, another cAMP-coupled PGE receptor, in NOTCH3-positive cells, the authors should evaluate potential contribution of EP4 receptor signaling to PGE2-induced effect.

CAY10684, an agonist of EP4, increased the cAMP level in Hu5/KD3 cells, but did not reduce the fusion index (**Fig. 7**). In addition, an EP4 antagonist, ONO-AE3-208, did not significantly increase the fusion index of Hu5/KD3 myogenic cells (**Fig. 7**) and hiPSC-derived muscle progenitors (**data not shown**), suggesting that the contribution of EP4 to PGE2-induced effects downstream Notch signaling is relatively small.

Moreover, the authors should show whether EP2 receptor is functionally expressed in muscle progenitor cells: PGE2 as well as butaprost (and also an EP4 agonist) is able to increase cAMP levels? Furthermore, such a cAMP-producing activities of PGE2 and butaprost were not detected in DAPT-treated cells?

The cAMP level was elevated by PGE2, butaprost, or CAY10684. These data indicate that both EP2 and EP4 receptors are functionally expressed in muscle progenitors. The cAMP-producing activities of PGE2 and butaprost were slightly reduced in the cells treated with DAPT for 3 days, although there was no statistically significant difference due to the high variations in the values (**Fig. 7, supplementary Fig. 9**). Importantly, CAY10684 increased the intracellular cAMP level as butaprost did, but the fusion index of CAY10684-treated cells was not reduced, suggesting that cAMP might not be the mediator for PGE2-induced self-renewal-promoting activities.

3) Lack of verification of cAMP and downstream pathways:

The authors discussed EP2-derived cAMP may participate in self-renewal of muscle progenitor cells (page 5, line 3 from bottom). If so, the effect of PGE2 or butaprost should be mimicked by membrane-permeable cAMP such as dibutyryl cAMP (dbcAMP). Conversely, the effects of PGE2 and butaprost are inhibited by H-89, an PKA inhibitor?

We tested the effects of dbcAMP and H-89, a PKA inhibitor, on the differentiation of myogenic cells. Unexpectedly, dbcAMP stimulated the differentiation of muscle progenitors. H-89 did not improve the differentiation of myogenic cells. We confirmed the results using both Hu5/KD3 myogenic cells and hiPSC-derived muscle progenitors (**Fig. 7 and supplementary Fig. 6**). These results suggest that the cAMP-PKA pathway does not regulate self-renewal of muscle progenitors downstream of Notch signaling.

**The effect of DAPT is reversed by PGE2, butaprost, an EP4 agonist, and dbcAMP?
Furthermore, even in the absence of NOTCH3, is dbcAMP able to maintain the self-renewal state in the muscle progenitors?**

The effects of DAPT were not reversed by butaprost in Hu5/KD3 cells, probably because DAPT down-regulated EP2. Only a high dose of PGE2 reversed the effects to some extent (data not shown).

CAY10684, an EP4 agonist, increased the cAMP concentration but did not inhibit fusion of myogenic cells (**Fig. 7**). In addition, dbcAMP promoted differentiation of Hu5/KD3 myogenic cells and hiPSC-derived muscle progenitors. H89, an inhibitor of PKA, did not promote the differentiation of Hu5/KD3 and hiPSC-derived muscle progenitors. Collectively, these results suggest that the Notch/EP2-mediated inhibition of muscle differentiation is not regulated by the cAMP-PKA signal pathway.

cAMP did not seem to mediate EP2 signaling for self-renewal of muscle progenitors (**Fig.7**). Therefore, we tested the effects of selective inhibitors of Src, PI3K, p38 MAPK, EGFR, JNK, or GSK-3 β on differentiation of muscle progenitors because they have been reported to mediate EP2 signaling in tumorigenesis, inflammation, or neuroprotection (reviewed in Jiang et al., 2013; Sun & Li, 2018). Interestingly, an Src family inhibitor, A419259, promoted the fusion of muscle progenitors at low concentrations (1–10 nM), and antagonized the effects of butaprost (**Fig. 8, supplementary Fig.10**), suggesting that Src is one of major signaling molecules that inhibit muscle differentiation downstream of the EP2 receptor. LY294002 (PI3K inhibitor) showed mildly antagonistic effects against butaprost (**supplementary Fig. 10**). SP600125 (JNK inhibitor), erlotinib (EGFR inhibitor), and CHIR99021 (GSK-3 β inhibitor) had little or no effect on the muscle differentiation of human muscle progenitors (**supplementary Fig. 10**). p38 MAPK is a well-known kinase which promotes muscle differentiation (Lluis et al., Trends Cell Biol., 16: 36-44, 2006). As expected, SB203580, an inhibitor of p38 MAPK suppressed muscle differentiation (**supplementary Fig. 10**), and butaprost further inhibited differentiation of muscle progenitors (**Fig. 8**). We added these results to the revised Ms.

Lim et al. have reported that inhibition of c-Src stimulated muscle differentiation *via* p38 MAPK activation (Lim et al., 2007). Chun *et al.* reported that c-Src complexes with β -arrestin 1 and EP2 to mediate EP2 signaling (Chun et al., 2010). Casini *et al.* reported that inhibition of the SRC family kinases promoted the muscle differentiation of rhabdomyosarcoma cells by decreasing the cleaved form of NOTCH3, and by activating p38 MAPK (Casini et al., 2015). Collectively, these reports and our results suggest an interesting model in which EP2 activates c-Src, which in turn activates NOTCH3, and eventually suppresses p38 MAPK activity to inhibit differentiation of human muscle progenitors, although this model remains to be proven. Especially whether EP2 signaling can modify NOTCH3 signaling or not is an important question to be answered.

Minor points:

1. Through the manuscript, some parts of text are written in red, what does it mean?

After first submission of our manuscript, we were asked to follow the journal's policy. We revised our Ms according to the policy of *Communications Biology*, and the changed parts that were rewritten are in red. Now the part is in black.

2. Page 2, line 2, 'tert. butyl' should be 'tertial butyl'.

We correctly spelled it 'tertial butyl' instead 'tert. butyl'.

3. Page 2, line 6, 'prostaglandin' should be 'prostaglandin (PG)'.

We added "(PG)".

4. Page 5. Line 5, Fig. 2d should be Fig. 3d.

We corrected the figure number.

5. Page 6, line 13, the authors should cite reference of an EP2-antagonist, TG6-10-1 (Jiang, J., et al. PNAS 110: 3591, 2013).

We cited this paper in the revised Ms.

Reviewers' comments:

Reviewer #2 (Remarks to the Author):

The issues raised in my prior review have been addressed in a satisfactory manner.

Reviewer #3 (Remarks to the Author):

General comments:

The authors responded well to the comments raised by this reviewer and the quality of the experiments included in the revised manuscript became much better than the original manuscript. However, unfortunately, the authors failed to detect effective dose of PGE2 levels in the culture system and found that EP2 receptor inhibits differentiation via the signaling other than the cAMP/PKA pathway.

Major concerns:

1)

The authors investigated PGE2 levels in the culture media only at single time point, but this is not adequate. PGE2 may be transiently synthesized and the released PGE2 may be catabolized by inactivating enzyme. Indeed, the authors detected only 15 pg/ml (approx. 0.04 nM) PGE2, which is much less than its effective concentration (> 1 nM). Therefore, the authors should have examined the time-course analysis on PGE2 synthesis.

Moreover, if the muscle progenitor cells are surrounded by mucosal fluid, PGE2 may not be released into media (trapped in the fluid). To overcome this situation, the authors should measure PGE2 levels not only in the media, but also in the 'cells' (including mucosal fluid).

In any case, to this reviewer, there still exists fatal discrepancy in this manuscript. The authors failed to certify that effective dose of PGE2 is endogenously synthesized in this culture system.

2)

Another problem in this manuscript is weak conclusion. The authors concluded that the downstream signaling pathway of EP2 receptor is not cAMP-PKA pathway, but c-Src/PI3K/inhibition of p38-MAPK. However, they concluded this mechanism only from experiments using inhibitors. To this reviewer, if the PGE2-EP2 receptor activation really affects muscle differentiation via the mechanism other than cAMP/PKA pathway, the signaling pathway should be the crucial part of this manuscript. The authors should show that PGE2 as well as butaprost but not CAY10684 (an EP4 agonist) really activates c-Src and inhibits p38 MAPK in muscle progenitor cells.

Reviewer #3 (Remarks to the Author):

General comments:

The authors responded well to the comments raised by this reviewer and the quality of the experiments included in the revised manuscript became much better than the original manuscript. However, unfortunately, the authors failed to detect effective dose of PGE2 levels in the culture system and found that EP2 receptor inhibits differentiation via the signaling other than the cAMP/PKA pathway.

Major concerns:

1) The authors investigated PGE2 levels in the culture media only at single time point, but this is not adequate. PGE2 may be transiently synthesized and the released PGE2 may be catabolized by inactivating enzyme. Indeed, the authors detected only 15 pg/ml (approx. 0.04 nM) PGE2, which is much less than its effective concentration (> 1 nM). Therefore, the authors should have examined the time-course analysis on PGE2 synthesis.

Moreover, if the muscle progenitor cells are surrounded by mucosal fluid, PGE2 may not be released into media (trapped in the fluid). To overcome this situation, the authors should measure PGE2 levels not only in the media, but also in the 'cells' (including mucosal fluid).

In any case, to this reviewer, there still exists fatal discrepancy in this manuscript. The authors failed to certify that effective dose of PGE2 is endogenously synthesized in this culture system.

PGE2 levels increase with differentiation

We appreciate reviewer's comments. In the new experiments, we sampled not only culture media but also cells, and measured the PGE2 concentration at five timepoints

during muscle differentiation. Until day 4, PGE2 levels were very low, but when cells started to form myotubes, the PGE2 level increased up to 200 pg/mL (approx. 0.5 nM) (Supplementary Figure 8).

Although the concentration was less than 1 nM in the early phase of differentiation, COX inhibition experiments (Fig.7) and EP2 overexpression and knockdown experiments (Fig.5) suggest that an effective dose of PGE2 is endogenously synthesized in this culture system.

We found the upregulation of PGE2 in the later stage of differentiation. Therefore, we examined *COX1* and *COX2* mRNA levels in myoblasts and myotubes by RT-qPCR. Although there was no statistical significance, *COX1* was found to be expressed at higher levels in myotubes rather than in myoblasts. In contrast, *COX2* tended to be highly expressed in myoblasts. The results suggest that *COX2* in myoblasts produced PGE2 in an early stage of differentiation to regulate cell fate (this study). Later, *COX1* in myotubes would produce PGE2 to keep satellite cells in an undifferentiated state in their niche. The latter hypothesis needs further experiments.

Upregulation of EP2 is enough to inhibit differentiation of myogenic progenitors

To clarify the roles of EP2, we up-regulated the expression of EP2 in Hu5/KD3 myogenic cells using two different plasmid vectors. Interestingly, mild upregulation of EP2 inhibited differentiation of the cells (new Figure 5h-j and supplementary Figure 5d). In contrast, modest downregulation of EP2 by shRNA increased the fusion index (Fig. 5). The results indicate that EP2 is a key molecule to regulate self-renewal of myogenic cells. The results also suggest that the expression level of EP2 is tightly regulated and that even in a low concentration of prostaglandinE2, upregulation of EP2 is enough for signal transduction. Of course, there is no doubt about the requirement of PGE2 for EP2 signaling as shown by the experiments using cox inhibitors (Fig.7).

2) Another problem in this manuscript is weak conclusion. The authors concluded that the downstream signaling pathway of EP2 receptor is not cAMP-PKA pathway, but c-Src/PI3K/inhibition of p38-MAPK. However, they concluded this mechanism only from experiments using inhibitors. To this reviewer, if the PGE2-EP2 receptor activation really affects muscle differentiation via the mechanism other than cAMP/PKA pathway, the signaling pathway should be the crucial part of this manuscript. The authors should show that PGE2 as well as butaprost but not CAY10684 (an EP4 agonist) really activates c-Src and inhibits p38 MAPK in muscle progenitor cells.

cAMP-PKA signaling

We further tested the effects of three activators and one inhibitor of the AC-cAMP-PKA and AC-cAMP-Epac signaling pathways on muscle differentiation.

- (1) forskolin (activator of AC)
- (2) ESI-09 (inhibitor of Epac)
- (3) 8-bromo-cAMP (selective activator of protein kinase A)
- (4) 8-CPT-2Me-cAMP (selective activator of Epac)

Again, none of them showed significant effects on differentiation of Hu5/KD3 myogenic cells (new Figure 7, supplementary Figure 10), confirming our previous observation that cAMP, which is produced upon activation of EP2, is not the signaling molecule responsible for regulation of self-renewal of muscle progenitors by EP2 signaling.

EP2 and Src signaling

Although our previous data showed that Src signaling negatively regulated muscle differentiation and a Src inhibitor antagonized butaprost's effects, we did not detect phosphorylation of Src by butaprost or PGE2 by using Western blotting (data not shown) or a human phospho-kinase array (Abcam, ARY003B). The array also suggested that Src

family kinases (Src, Lyn, Lck, Fyn, Yes, Fgr, and Hck), ERK1/2, JNK1/2, EGFR, GSK-3 β , Akt, and CREB are not direct targets of EP2 signaling (please see attached figure). Therefore, we deleted the data on Src and PI3K from the manuscript. On a human phospho-kinase array, butaprost slightly reduced phosphorylation of p38 α -MAPK, a key signaling molecule for muscle differentiation. Western blotting also showed lower levels of phosphorylation of p38 α -MAPK in butaprost-treated cells than in control cells (please see attached figure). Therefore, p38 α -MAPK might be a target of EP2 signaling. In near future, we will clarify the molecules that link EP2 and p38 α MAPK.

Thus, how EP2 regulate self-renewal of muscle progenitors is not clear. We stated this limitation of our study in the Discussion.

figure for rebuttal

Phospho-kinase array and Western blotting show that p38 α -MAPK phosphorylation level is decreased by butaprost

- A) Experimental design. Hu5/KD3 cells were seeded at 1.2×10^6 cells/10 cm² collagen-coated dish. Proteins were extracted after two hours treatment with 0.1% DMSO, 10 μ M DAPT, 500 ng/mL butaprost or 10 μ M CAY10684.
- B) Images of phospho-kinase array membranes.
- C) Phosphorylation of Src family kinases and p38 α MAPK compared with control (0.1% DMSO-treatment).
- D) p38 α -MAPK and pp38 α -MAPK expression levels during muscle differentiation with or without butaprost treatment.

REVIEWERS' COMMENTS:

Reviewer #3 (Remarks to the Author):

The authors responded well to the concern at least regarding PGE2 synthesis and contribution of each COX isozyme during transition from myoblasts to myotubes. These results greatly improved the missing points in the previous manuscript. The authors failed to identify a novel EP2-mediated signaling pathway other than cAMP-PKA, and modified their conclusion in an appropriate manner. Therefore, the revised paper is now acceptable.